# Feed-forward loops by NR5A2 ensure robust gene activation during pre-implantation development

Wataru Kobayashi*, Siwat Ruangroengkulrith*, Eda Nur Arslantas, Adarsh Mohanan and Kikuë Tachibana‡

## ABSTRACT

Pioneer transcription factors are crucial for regulating zygotic genome activation and cell differentiation during mouse pre-implantation development. However, how pioneer factors function collectively to regulate early development remains poorly understood. Here, we determined the chromatin-binding profiles of the mouse pioneer factor NR5A2 during the totipotency-to-pluripotency transition and identified KLF and GATA family transcription factors as key co-regulators. NR5A2 regulates the expression of *Klf5* and *Gata6*, the protein products of which in turn act as co-regulators of NR5A2 to promote development. Mechanistically, KLF5 contributes to H3K27ac deposition at genomic regions co-occupied by NR5A2. NR5A2 also regulates *Xist* expression, either directly or indirectly, through its role in co-binding with GATA factors and upregulating their expression. *In vitro* assays revealed that NR5A2 binds to nucleosomes with KLF5 and GATA6, suggesting that these pioneer factors can simultaneously bind to chromatin. Our findings provide evidence for a feed-forward regulatory mechanism by which NR5A2 activates expression of lineage-determining factors and these, together with NR5A2, subsequently co-bind nucleosomes to ensure robust gene activation during pre-implantation development.

KEY WORDS: Lineage-determining factor, Nuclear receptor, Pioneer transcription factor, Pre-implantation development, Transcriptional regulation, Mouse

## INTRODUCTION

Upon fertilization, the zygote acquires totipotency, which is the developmental potential to generate all cell types and form a complete organism. In mice, totipotency is maintained through the zygote and 2-cell stages (Tarkowski, 1959), and the potential of totipotency gradually decreases during the cleavage divisions until reaching a pluripotent or differentiated state (Rossant, 1976; Kelly, 1977). Initially, transcriptionally silent embryos are 'awakened' by a process referred to as zygotic genome activation (ZGA) (Jukam et al., 2017; Kravchenko and Tachibana, 2025). Following ZGA, embryos undergo cell compaction at the 8-cell stage and establish cell polarity. At the morula stage, inner and outer cells with different polarization states influence the Hippo signaling pathway, which in

turn determines their cell fate. Upon reaching the blastocyst stage, inner and outer cells form the inner cell mass (ICM) and the trophectoderm (TE), respectively (Rossant and Tam, 2009; Zernicka-Goetz et al., 2009; Rossant, 2018). The ICM subsequently gives rise to the pluripotent epiblast and the primitive endoderm (PrE) (Rossant and Tam, 2009; Zernicka-Goetz et al., 2009; Rossant, 2018). These developmental transitions are driven by dynamic transcriptional regulatory networks that govern cell fate specification during pre-implantation development.

Pioneer transcription factors (TFs) play crucial roles in facilitating chromatin remodeling and driving cell fate transitions (Zaret and Carroll, 2011; Iwafuchi-Doi and Zaret, 2014; Barral and Zaret, 2024). Among these, Krüppel-like factor (KLF) and GATA TFs are known as regulators of early embryonic development. The KLF and GATA families consist of 17 and six members, respectively, with both overlapping and distinct roles. For example, KLF17, a maternally derived factor, is required for ZGA at the 2-cell stage (Hu et al., 2024), whereas KLF4 and KLF5 are expressed during ZGA and contribute to lineage specification towards ICM and TE (Ema et al., 2008; Lin et al., 2010; Kinisu et al., 2021). GATA4 and GATA6 are known as PrE lineage markers (Morrisey et al., 1998; Koutsourakis et al., 1999), while GATA3 contributes to TE lineage specification (Home et al., 2009; Ralston et al., 2010; Gerri et al., 2020). The highly conserved DNA-binding domain within these families suggests both functional redundancy and cell stage-specific functions. However, the detailed regulatory mechanisms underlying pioneer factor functions at mammalian pre-implantation stages remain poorly understood.

The orphan nuclear receptor NR5A2 is a pioneer TF that is essential for pre-implantation development in mice (Gassler et al., 2022; Lai et al., 2023; Festuccia et al., 2024; Kobayashi et al., 2024; Li et al., 2024; Zhao et al., 2024). NR5A2 regulates ZGA in the 2-cell embryo (Gassler et al., 2022) and controls expression of lineage-determining factors at the 8-cell stage (Lai et al., 2023; Festuccia et al., 2024; Li et al., 2024; Zhao et al., 2024). *Nr5a2* knockout or knockdown (KD) embryos arrest at the morula stage, demonstrating that NR5A2 is a pivotal factor that ensures proper pre-implantation development. Thus, NR5A2 orchestrates distinct transcriptional regulatory networks during the totipotency-to-pluripotency transition. However, the mechanisms by which NR5A2 differentially regulates transcriptional networks at distinct developmental stages remain unknown. Given that eukaryotic regulatory elements are often complex and involve the action of multiple TFs, we hypothesized that NR5A2 functions with lineage-determining TFs to control distinct transcriptional networks during pre-implantation development.

## RESULTS

### NR5A2 binding during the totipotency-to-pluripotency transition

To investigate the transcriptional networks regulated by NR5A2, we first asked where NR5A2 is bound in 2-cell, 4-cell, 8-cell and morula-stage embryos. Using CUT&Tag (Kaya-Okur et al., 2019)

Department of Totipotency, Max Planck Institute of Biochemistry (MPIB), 82152, Munich, Germany.
*These authors contributed equally to this work

‡Author for correspondence (tachibana@biochem.mpg.de)

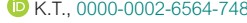 K.T., 0000-0002-6564-7484

**DEVELOPMENT**

of approximately 800 cells per stage, genome-wide NR5A2 binding profiles were determined (Fig. 1A, Fig. S1A). As a reference, previously published ChIP-seq data (Atlasi et al., 2019; GSE92412) in embryonic stem cells (ESCs) are shown. *De novo* motif analysis showed that the peaks are enriched for the NR5A2 motif in each cell stage (Fig. S1B). Our analysis detected 5836, 9251, 19,839 and 14,854 reproducible peaks in 2-cell, 4-cell, 8-cell and morula-stage

embryos, respectively (Table S1). These data suggest that NR5A2 binds the genome most extensively at the 8-cell stage, and binding sites decrease at the morula stage. Alluvial plots showed a gain and loss of NR5A2 peaks during the cell-stage transitions (Fig. 1B). Although we cannot exclude that non-peaks could be a false negative due to the small cell numbers, the overall picture emerges that NR5A2 dynamically changes its chromatin binding with each cell cycle.

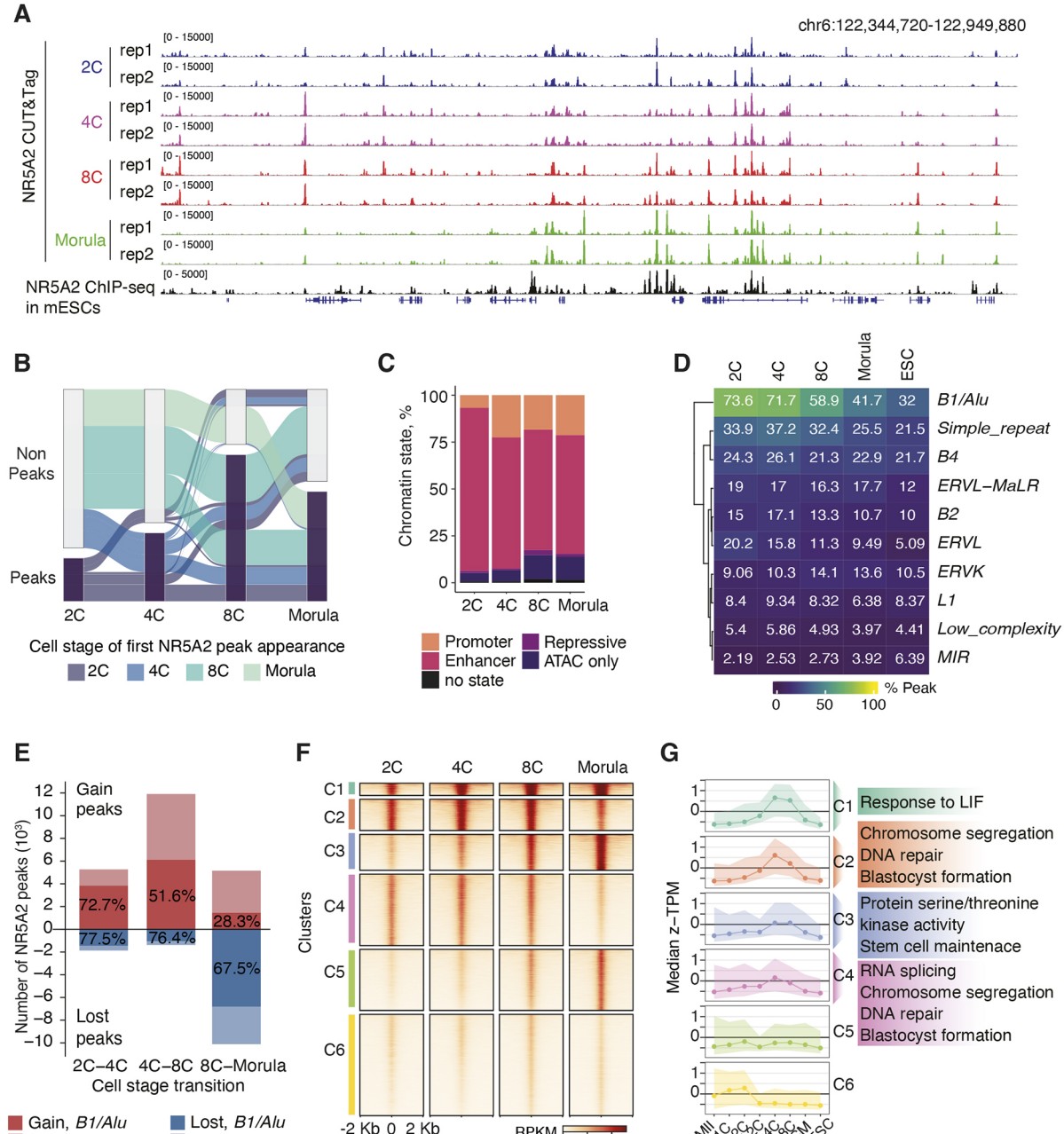

**Fig. 1. NR5A2 chromatin binding dynamics from the 2-cell to morula stage.** (A) Integrated Genome Viewer (IGV) snapshot showing NR5A2 CUT&Tag signals across developmental stages: 2-cell (34-36 hpf, blue), 4-cell (41-43 hpf, purple), 8-cell (56-58 hpf, red) and morula (71-72 hpf, green), with two biological replicates. NR5A2 ChIP-seq data in mouse embryonic stem cells (mESCs, black) is shown for reference. (B) Alluvial plot illustrating the dynamic changes in NR5A2 chromatin binding at each cell stage transition. (C) Classification of chromatin states associated with NR5A2 binding at each cell stage. Chromatin states were identified using ChromHMM. (D) Heatmap depicting the enrichment of repetitive elements (subfamilies) within NR5A2 peaks. The percentage indicates the proportion of NR5A2 peaks that overlap with repetitive elements, including cases where multiple elements are present within a single peak. (E) Bar chart representing the gain (red) and loss (blue) of NR5A2 peaks in each developmental transition. The percentage indicates the proportion of *SINEB1/Alu* peaks. (F) Heatmap of NR5A2 binding transition from the 2-cell to the morula stage. The binding loci were clustered by k-mean clustering. (G) Stage-specific expression levels of genes associated with each NR5A2-binding site cluster (left) and their associated gene ontology terms (right). 1C, zygote; 2C, 2-cell stage; 4C, 4-cell stage; 8C, 8-cell stage; e2C, early 2-cell; MII, metaphase II.

NR5A2 predominantly targets putative enhancer regions including the retrotransposon element *SINE B1/Alu* at the 2-cell stage when ZGA occurs (Gassler et al., 2022; Kravchenko and Tachibana, 2025). To investigate chromatin states targeted by NR5A2 at each cell stage, we applied a hidden Markov model (HMM) to infer putative *cis*-regulatory elements from chromatin accessibility and histone modifications using ChromHMM (Ernst and Kellis, 2017). To this end, H3K27ac CUT&Tag and omni ATAC-seq were conducted, and the profiles of H3K4me3, H3K9me3 and H3K27me3 were obtained from public datasets (Fig. S1C-F, Table S2; GSE72784, GSE73952, GSE71434 and GSE98149). Our analysis showed that NR5A2 preferentially targets promoters and putative enhancers from the 2-cell stage to the morula stage, which implies that NR5A2 is required for transcriptional activation (Fig. 1C). Consistent with our previous report (Gassler et al., 2022), 73.6% NR5A2 peaks overlapped with *SINE B1/Alu* at the 2-cell stage. Intriguingly, the proportion of *SINE B1/Alu* in NR5A2 peaks decreased gradually from the 8-cell stage, reaching 42% in the morula and 32% in ESCs (Fig. 1D). NR5A2 peaks at *SINE B1/Alu* were greatly increased at the 2-cell-to-4-cell and the 4-cell-to-8-cell transition (Fig. 1E). In contrast, during the 8-cell-to-morula transition, the gain of NR5A2 peaks at *SINE B1/Alu* was decreased, accompanied by a pronounced loss of NR5A2 peaks at *SINE B1/Alu* (Fig. 1E). These results suggest that NR5A2 specifically targets the retrotransposable elements *SINE B1/Alu* from the 2-cell until the 8-cell stage, which corresponds to the intermediate of the totipotency-pluripotency transition.

To understand whether binding of NR5A2 correlated with expression of the nearby genes, we first classified all NR5A2-binding sites based on the change of binding signals (Fig. 1F). The binding profile of NR5A2 was similar between 2-cell and 4-cell stages, before shifting drastically at the morula stage. The 8-cell stage appeared as a transition stage, consistent with the highest number of peaks detected. The k-mean clustering revealed four binding patterns of NR5A2 before the first lineage segregation (Fig. 1F). These were: (1) constant NR5A2-binding sites [cluster 1, (C1)], (2) binding sites from 2-cell to 8-cell stages (C2 and C4), (3) loci that gradually gained binding in later stages (C3 and C5), and (4) weak binding (C6). We next associated genes containing NR5A2 peaks within 10 kb of one of these binding clusters (Fig. 1G). We observed that a strong enrichment of NR5A2 binding is correlated with elevated gene expression at 4-cell and 8-cell stages (C1-C4). In contrast, clusters with weak NR5A2 enrichments (C6) did not show clear stage-specific gene expression. The binding sites gained from 8-cell to morula stage (C5) showed little to no association with gene expression, which could be consistent with these NR5A2-binding sites functioning as distal enhancers, potentially regulating target genes over long genomic distances.

Gene ontology analysis showed that constant NR5A2-binding sites were proximal to genes responding to leukemia inhibitory factor (LIF) (Fig. 1G, C1). NR5A2 binding from the 2-cell to the 8-cell stages was enriched near genes essential for chromosome segregation, DNA repair and blastocyst formation (Fig. 1G, C2 and C4). Notably, a cluster of NR5A2 binding, which peaked at the morula stage, was found near genes related to stem cell maintenance and protein serine/threonine kinase activity, including a component of Hippo signaling pathway and polarization, *Lats2*, and atypical protein kinase C (*Prkcz*) (Fig. 1G, C3). This raises the possibility that NR5A2 contributes to the first lineage specification by promoting the Hippo signaling pathway.

## KLF5 and GATA6 with NR5A2 co-occupy active chromatin

To identify potential co-regulators for NR5A2 in different cell stages, we investigated TF motifs enriched in NR5A2 peaks. The OCT4-SOX2 motifs are highly enriched in ESCs (Fig. 2A), suggesting that

NR5A2 functions with OCT4 (POU5F1) and SOX2 in pluripotent cells (Festuccia et al., 2021). However, the OCT4-SOX2 motifs are not enriched in NR5A2 peaks from the 2-cell to the morula stage. Instead, our *de novo* motif searches identified KLF and GATA motifs in NR5A2 peaks and revealed that the motif enrichment increased at the 8-cell and morula stages (Fig. 2A). KLF motifs had also been identified in NR5A2 peaks at the 8-cell stage previously (Festuccia et al., 2024). We then examined a public ribosome profiling (Ribo-seq) dataset (Xiong et al., 2022; GSE165782) and found that KLF5 and GATA6 are upregulated from the 8-cell to blastocyst stage among these family members (Fig. 2B). KLF5 and GATA6 are known as lineage-determining factors in mouse pre-implantation development. KLF5 is required to establish bi-potential cell fate for both ICM and TE gene expression (Ema et al., 2008; Lin et al., 2010; Kinisu et al., 2021), while GATA6 plays a crucial role in PrE differentiation at the blastocyst stage (Morrisey et al., 1998; Koutsourakis et al., 1999; Wamaitha et al., 2015). These results indicate that NR5A2 may function with lineage-determining factors after ZGA.

To test whether NR5A2 co-occupies chromatin with these lineage-determining factors, we performed CUT&Tag for KLF5 and GATA6 at the 8-cell and morula stages. As expected, KLF5 and GATA6 peaks were detected at a subset of NR5A2-binding sites, suggesting that these TFs co-occupy chromatin at specific genomic loci (Fig. 2C,D). *De novo* motif analysis showed that canonical KLF5 and GATA motifs are enriched within these peaks (Fig. S2A). Principal component analysis showed the trajectory of NR5A2 chromatin binding, which follows the order of development. Interestingly, NR5A2 binding profiles at the morula stage tended to be similar to those of KLF5 and GATA6 at the 8-cell stage (Fig. S2B). To determine whether these co-occupied sites correlate with NR5A2 function at a later development stage, we classified NR5A2 peaks at the morula stage into four clusters based on signal of the three factors (Fig. 2E). Overall, the progressive increase of NR5A2 occupancy from the 2-cell to morula stages was observed in all clusters. Clusters 1 and 2 are loci that were strongly bound by NR5A2 from the 2-cell stage, while clusters 3 and 4 showed little to no NR5A2 occupancy at early stages. We identified strong occupancies of KLF5 and GATA6 at the morula stage at binding sites from clusters 1 and 3, suggesting that these sites are co-occupied by all three TFs. Notably, the high occupancy of these factors also correlated with a marked increase in H3K27ac at the morula stage (Fig. 2E, clusters C1 and C3). Consistent with earlier results (Fig. 1D), loci that were constantly bound by NR5A2 (cluster 2) contained the highest density of *SINE B1/Alu*. In contrast, newly gained binding sites (clusters 3 and 4) preferentially located at non-*SINE B1/Alu* regions with a somewhat degenerated NR5A2 motif (Fig. 2E, Fig. S2C,D). These data suggest that the potential co-occupancy of these three TFs is associated with the establishment of active promoters or enhancers.

## NR5A2 and KLF5 function synergistically in embryonic development

To investigate whether NR5A2 regulates gene expression alongside KLF5 or GATA6 during early embryonic development, we performed small interfering RNA (siRNA)-mediated KD in zygotes to perturb their functions. We found that *Gata6* KD was inefficient up to the 8-cell stage (Fig. S3A), requiring more than three cell divisions for protein depletion. Therefore, we focused on cooperativity of KLF5 with NR5A2. To deplete both NR5A2 and KLF5 proteins, siRNAs targeting *Nr5a2* and *Klf5* and co-injection marker *H2B-GFP* mRNA were microinjected into zygotes and cultured until the 8-cell stage (Fig. 3A). The 8-cell embryos were subsequently subjected to immunofluorescence analysis and RNA-seq. KD of either *Nr5a2* or

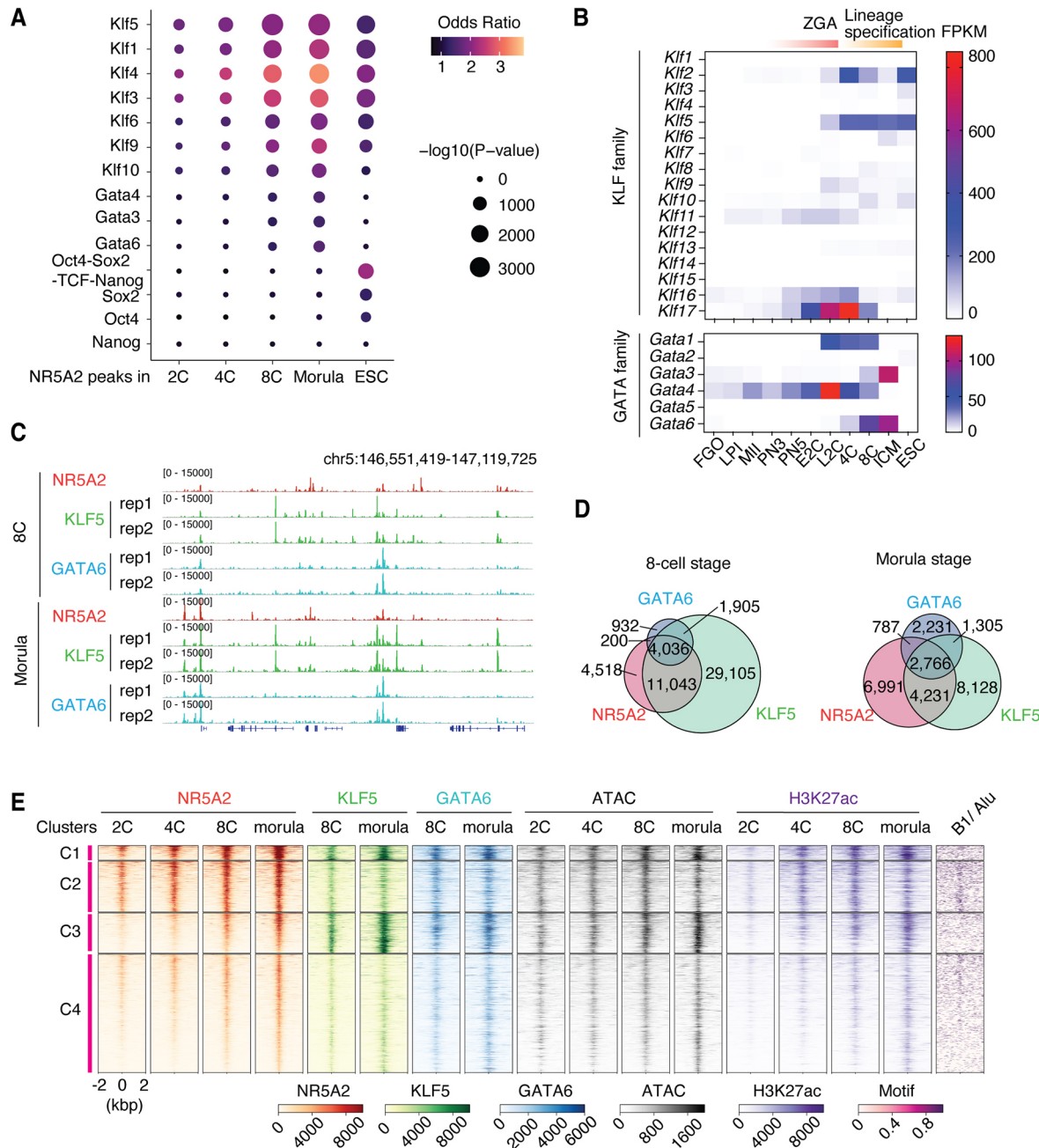

**Fig. 2. Identification of co-regulators for NR5A2.** (A) Co-enrichment of TF motifs with NR5A2 peaks in each cell stage. The color indicates the odds ratio between observed and expected motif occurrence within the peak. The size of circle indicates the *P*-value. (B) Ribo-seq data (Xiong et al., 2022) showing the expression levels of KLF and GATA families at different developmental stages. CUT&Tag profiles for KLF5 and GATA6 with two replicates at the 8-cell and morula stages are shown. (D) Venn diagrams showing the number of overlapping peaks between NR5A2, KLF5 and GATA6 at the 8-cell and the morula stages. (E) Heatmap showing the enrichment of NR5A2, KLF5, GATA6, ATAC-seq and H3K27ac from the 2-cell to the morula stage with four different clusters. *SINE B1/Alu* motif occurrence is shown on the right. All maps extend for 2 kb on either side. 4C, 4-cell stage; 8C, 8-cell stage; E2C, early 2-cell embryo; FGO, full-grown oocytes; ICM, inner cell mass; L2C, late 2-cell embryo; LPI, late prometaphase I oocytes; MII, metaphase II oocytes; PN, pronuclear stage. (C) IGV snapshot showing co-occupancy of KLF5 (green) and GATA6 (blue) with NR5A2 (red).

*Klf5* resulted in efficient depletion of the respective mRNA and protein by the 8-cell stage (Fig. 3B-D). Notably, *Klf5* mRNA and protein abundances were also reduced upon *Nr5a2* KD, suggesting that *Klf5* expression is regulated by NR5A2 (Fig. 3B,C). This is consistent with our previous finding that NR5A2 inhibition causes a significant downregulation of *Klf5* transcription in 2-cell embryos (Gassler et al., 2022). Double KD of *Nr5a2 Klf5* (dKD) led to a further reduction of *Klf5* mRNA compared to *Nr5a2* KD alone and efficiently depleted both NR5A2 and KLF5 proteins (Fig. 3B-D).

Next, we tested whether NR5A2 and KLF5, alone or in combination, are required for development. Under our culture conditions, control embryos developed to the blastocyst stage, with a hatching rate (defined as breaking out of the zona pellucida) of >60% at 108 h post-fertilization (hpf) (Fig. 3E,F). *Nr5a2* KD embryos developed with a delay and fragmented around the morula stage, consistent with previous work (Fig. 3E,F) (Lai et al., 2023; Festuccia et al., 2024; Zhao et al., 2024). In contrast, *Klf5* KD resulted in a mild developmental defect and blastocysts contained a slightly smaller

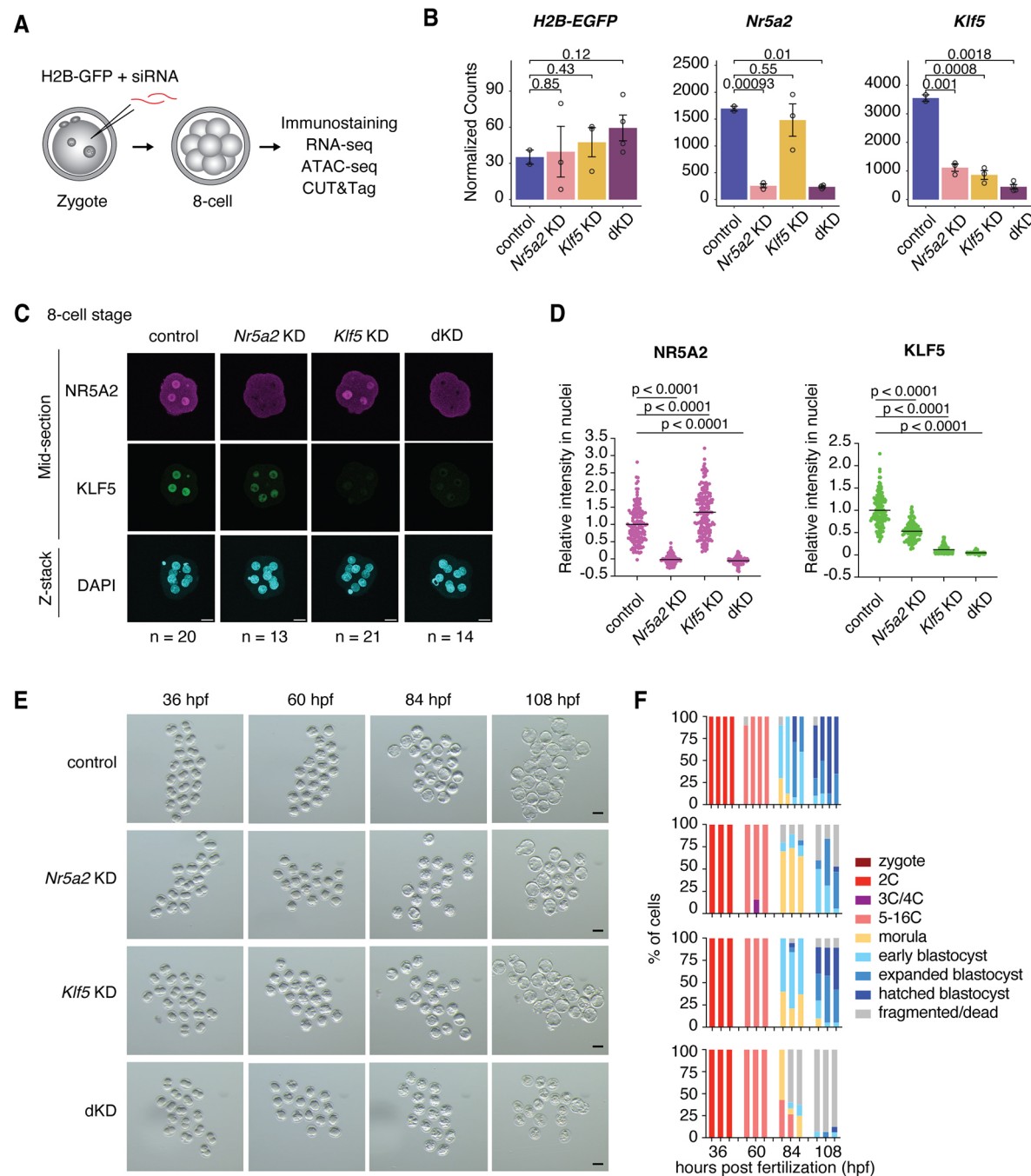

**Fig. 3. NR5A2 and KLF5 contribution to pre-implantation development.** (A) Schematic of siRNA-mediated knockdown. siRNA targeting *Nr5a2* and/or *Klf5* was microinjected into zygotes. Immunostaining and genomics assays were performed with 8-cell embryos. (B) Abundance of co-injection marker *H2B-EGFP*, *Nr5a2* and *Klf5* transcripts in control (blue), *Nr5a2* KD (pink), *Klf5* KD (yellow) and *Nr5a2 Klf5* dKD (purple) 8-cell embryos. *P*-values (*t*-test, two-sided) are shown. Each dot represents a replicate of the RNA-seq experiment. Error bars represent s.d. (C,D) Representative images (C) and quantification (D) of immunostaining analysis showing NR5A2 (magenta), KLF5 (green) and DAPI (cyan) staining in 8-cell embryos. NR5A2 and KLF5 signals are shown in mid-section images. DAPI signals are presented as full *z*-stack images to visualize all nuclei. The number of embryos examined (*n*) from three independent experiments is indicated. Scale bars: 20 μm. Bars overlaid on the plots indicate means. *P*-values (*t*-test, two-sided) are shown. (E) Stereomicroscopic representative images showing embryonic development at different time points under different KD conditions (top to bottom: control, *Nr5a2* KD, *Klf5* KD, dKD). Scale bars: 75 μm. (F) Quantification of embryonic development experiments. The number of embryos examined (*n*) from three or four independent experiments were as follows: control, *n*=10, 8, 24 and 20; *Nr5a2* KD, *n*=10, 19 and 17; *Klf5* KD, *n*=10, 19 and 19; *Nr5a2 Klf5* dKD, *n*=14, 15 and 16.

blastocoel (Fig. 3E,F). Remarkably, *Nr5a2 Klf5* double KD caused a severe developmental failure, with >90% of embryos dying and fragmenting (Fig. 3E,F). Immunofluorescence analysis showed that protein abundances of the first lineage markers NANOG (ICM

marker) and CDX2 (TE marker) were significantly decreased in dKD embryos (Fig. S3B,C), indicating defects in the first lineage segregation. These results suggest that NR5A2 and KLF5 function synergistically in embryonic development.

## KLF5 contributes to H3K27ac deposition at genomic regions co-occupied by NR5A2

We next examined genome-wide transcriptional changes regulated by NR5A2 and KLF5 at the 8-cell stage (Fig. S4A, Table S3). *Nr5a2* KD and dKD with *Klf5* resulted in the downregulation of 483 and 489 genes, respectively, with 354 genes commonly affected in both conditions (Fig. 4A,B). In contrast, *Klf5* KD resulted in downregulation of only 29 genes (Fig. 4A,B). We cannot exclude the possibility that the minor effects of *Klf5* KD alone are due to inefficient knockdown (Fig. 3B-D). Downregulated genes upon *Nr5a2* KD and dKD were more enriched for NR5A2 and KLF5 peaks in their neighboring distal regions compared with upregulated genes (Fig. S4B), supporting a direct activation role for NR5A2 and KLF5. In addition, both *SINE B1/Alu*- and non-*SINE B1/Alu*-bound NR5A2 peaks also preferentially located near downregulated genes upon dKD (Fig. S4C). These results suggest that NR5A2 regulates gene expression through binding to both *SINE B1/Alu* and non-*SINE B1/Alu* regions.

The genes downregulated upon *Nr5a2* KD include early ICM and TE genes (Fig. S4D, Table S3), consistent with previous findings (Lai et al., 2023; Festuccia et al., 2024). We found that *Nr5a2* KD resulted in the downregulation of genes functioning in the Hippo signaling pathway, such as *Lats2* and *Prkcz*, suggesting a role for NR5A2 in the first lineage specification. NR5A2 regulates 74% (360 out of 489) of the genes downregulated in dKD, while an additional 128 genes were specifically downregulated in dKD (Fig. 4B). These dKD-specific downregulated genes are involved in categories related to Wnt/TGFβ signaling (*Prkcd*, *Smad7* and *Ryk*) and DNA repair (*Rad51c*, *RNaseh2c*, *Pold1* and *Aunip*), potentially explaining the severe developmental defects in the dKD condition (Fig. 4B).

To elucidate the mechanism by which NR5A2 and KLF5 regulate chromatin accessibility and/or promoter/enhancer activity, we performed ATAC-seq (Corces et al., 2017) and H3K27ac CUT&Tag in each KD condition at the 8-cell stage (Fig. 3A, Fig. S5A,B). Differential binding analysis revealed a strong depletion of H3K27ac in both *Nr5a2* KD and dKD conditions (Fig. 4C, Fig. S5C). Although our ATAC-seq data could not be used for this analysis due to variability in signal detection in the control condition, chromatin accessibility showed similar trends in regions where H3K27ac is affected (Fig. S5D). We observed a reproducible reduction of chromatin accessibility on NR5A2-binding sites near downregulated genes by *Nr5a2* KD (Fig. S5F), in agreement with previous reports (Lai et al., 2023; Festuccia et al., 2024). These data support a function of NR5A2 as a pioneer TF that promotes chromatin accessibility and suggest an involvement in the deposition of H3K27ac at regulatory elements.

In contrast, *Klf5* KD had negligible effects on H3K27ac deposition (Fig. 4C, Fig. S5C), consistent with the mild effects observed by RNA-seq. Interestingly, we observed more H3K27ac affected regions in *Nr5a2 Klf5* dKD than in the single *Nr5a2* KD condition (Fig. 4C, Fig. S5D). Moreover, regions with a strong reduction of H3K27ac tended to be co-bound by both NR5A2 and KLF5 (Fig. 4D, Fig. S5E), implying synergistic effects of NR5A2 and KLF5. Integrative Genomics Viewer (IGV) snapshots showed that *Nr5a2* KD and dKD led to reduced H3K27ac at NR5A2 binding sites near the early ICM-, TE- and Hippo signaling-related genes (Fig. 4E). A marked reduction of H3K27ac was observed at genes specifically downregulated in *Nr5a2 Klf5* dKD (Fig. 4E). Collectively, these results indicate that KLF5 contributes to H3K27ac deposition at genomic regions co-occupied by NR5A2 to regulate gene expression.

To test the proximity of NR5A2 and KLF5 in individual cells of an embryo, we performed a proximity ligation assay (PLA) (Soderberg et al., 2006). This method detects the close physical proximity of two target proteins, where oligonucleotide-conjugated antibodies in proximity generate a quantifiable rolling-circle amplification signal. PLA signals were detected significantly above background only in the presence of both NR5A2 and KLF5 antibodies, suggesting that these TFs are in close association in the nuclei of 8-cell stage embryos (Fig. 4F,G). The results of this orthogonal approach are consistent with the co-occupancy of TFs determined by CUT&Tag (Fig. 2D).

We next investigated how NR5A2 and KLF5 regulate each other. We found that NR5A2 binds to *Klf5*-adjacent distal regulatory elements that control *Klf5* expression in ESCs (Su et al., 2025). *Nr5a2* KD led to reduced H3K27ac at this regulatory region, supporting a role for NR5A2 in regulating *Klf5* expression (Fig. S6A). We then tested whether KLF5 promotes NR5A2 chromatin binding. To increase the sensitivity of chromatin binding profiles with a limited number of input cells, targeted insertion of promoters sequencing (TIP-seq) (Bartlett et al., 2021) was performed for NR5A2 in control and *Klf5* KD 8-cell embryos (Fig. S6B). NR5A2 occupancy was not significantly altered upon *Klf5* KD (Fig. S6C), indicating that NR5A2 chromatin binding is largely independent of KLF5.

## NR5A2 regulates GATA factors and *Xist*

In early embryonic development, GATA TFs regulate the expression of *Xist*, a non-coding RNA essential for initiating X chromosome inactivation (XCI) (Ravid Lustig et al., 2023). Our RNA-seq data revealed that *Nr5a2* KD resulted in downregulation of *Xist* and its regulator *Gata1* (Fig. 5A). While *Gata4* and *Gata6* showed only mild transcriptional downregulation, immunofluorescence analysis showed a significant reduction in GATA6 protein upon *Nr5a2* KD (Fig. 5B,C). NR5A2 binds to the *Gata1* promoter and distal regions near the *Gata6* transcription start site (TSS), and *Nr5a2* KD led to reduced H3K27ac at these regions (Fig. S7A). These data indicate that NR5A2 acts upstream of these GATA TFs and is associated with their expression. To exclude the possibility that our observed effect on XCI genes resulted from the stochastic imbalance of the ratio between male and female embryos in bulk RNA-seq, we re-analyzed published single-embryo RNA-seq data (Festuccia et al., 2024) with *Nr5a2* maternal-zygotic knockout 8-cell embryos. We found that *Xist* is substantially decreased in female embryos (Fig. S7B,C). These findings raise the interesting possibility that NR5A2 functions upstream of GATA TFs in regulating imprinted XCI.

Previous studies identified distal regulatory elements (REs) 79 and 97 as *Xist* enhancers targeted by GATA TFs (Ravid Lustig et al., 2023). We mapped GATA1 chromatin binding profiles using CUT&RUN (Fig. S8) (Skene and Henikoff, 2017) because data obtained by CUT&Tag was relatively sparse data for GATA1. We observed GATA6 and slightly weak GATA1 occupancy at RE79 and RE97 at the 8-cell stage (Fig. 5D), supporting a role of GATA factors in targeting these distal REs to regulate imprinted XCI in pre-implantation development (Ravid Lustig et al., 2023). Interestingly, we found that NR5A2 co-occupies RE79 with GATA1 and GATA6, but is not detected at RE97. In addition to these REs, NR5A2 and GATA6 co-occupancy was also detected at RE78 (Gjaltema et al., 2022), a locus where GATA6 is not bound in extra-embryonic endoderm cells (Fig. 5D). Although it remains to be tested whether RE78 directly regulates *Xist*, these data raise the possibility that NR5A2 regulates *Xist* expression, either directly or indirectly, through its role in upregulating GATA factor expression.

## NR5A2 co-binds to nucleosomes with KLF5 and GATA6 *in vitro*

Our genomics analyses suggested that NR5A2 co-occupies chromatin with KLF5 and GATA6 to regulate transcriptional

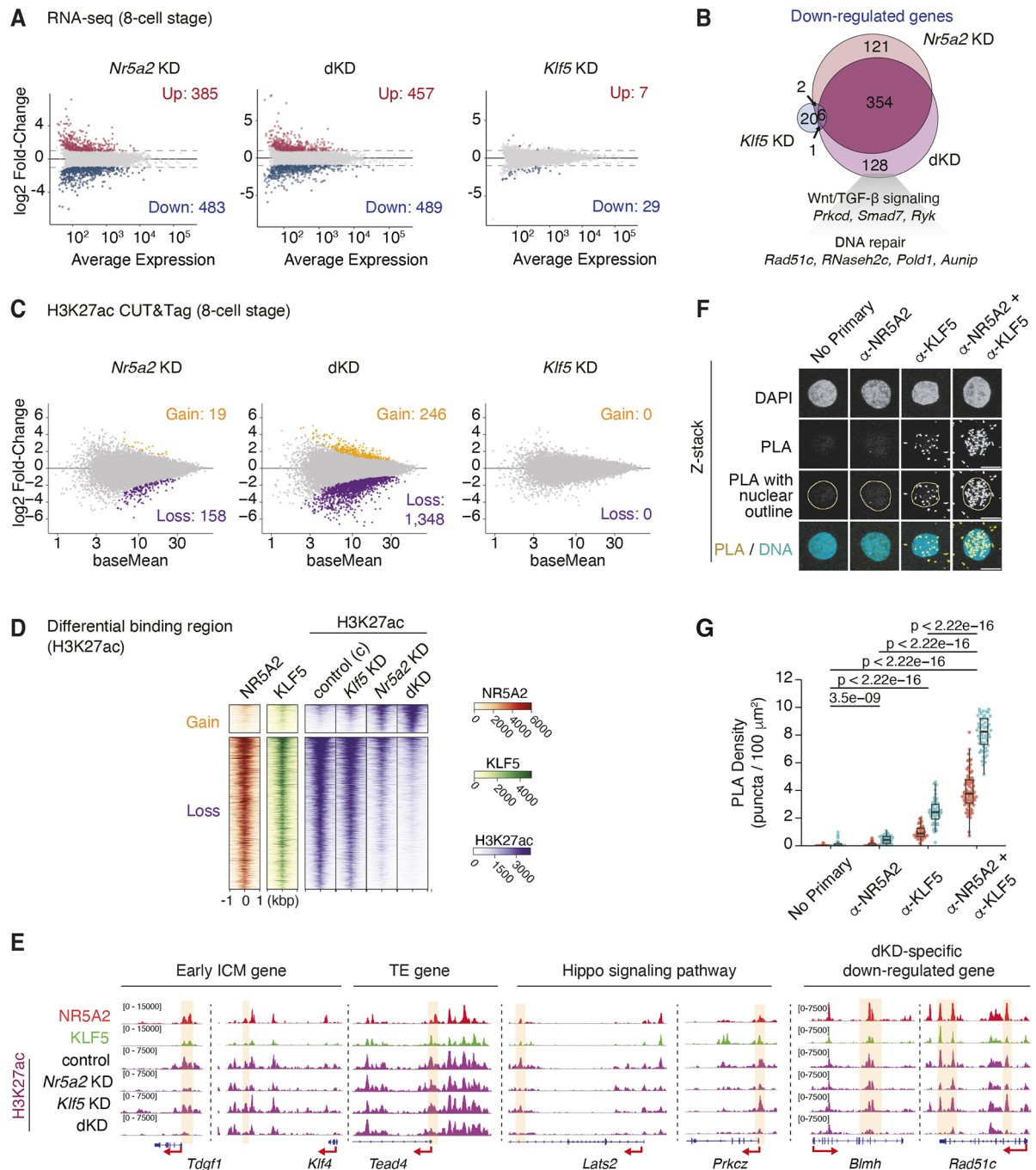

**Fig. 4. Transcriptional regulation by NR5A2 and KLF5 at the 8-cell stage.** (A) MA plots of the log$_2$ fold-change (*Nr5a2* KD/control, dKD/control and *Klf5* KD/control) in gene expression in *Nr5a2* KD (left), dKD (center) and *Klf5* KD (right). The number of up- and downregulated genes in each condition is shown. (B) Venn diagram showing the number of downregulated genes overlapping in *Nr5a2* KD, dKD and *Klf5* KD. The dKD-specific downregulated category contains genes related to Wnt/TGFβ signaling (*Prkcd*, *Smad7* and *Ryk*) and DNA repair (*Rad51c*, *Rnaseh2c*, *Pold1* and *Aunip*). (C) Differential binding analysis of H3K27ac CUT&Tag in *Nr5a2* KD (left), dKD (center) and *Klf5* KD (right). The numbers indicate gain and loss of H3K27ac regions in each condition. (D) Heatmap showing enrichment of H3K27ac on gain and loss of H3K27ac regions identified by differential binding analysis. H3K27 CUT&Tag data from three biological replicates merged are shown in each condition. (E) IGV snapshot highlighting the representative early ICM, TE, Hippo signaling pathway genes, and dKD-specific downregulated genes. H3K27ac CUT&Tag data are shown as merged profiles. (F) PLA showing a close association between NR5A2 and KLF5 in the nucleus. Eight-cell embryos were incubated with NR5A2 and/or KLF5 antibodies, and PLA signals were detected using a confocal microscope. A representative nucleus with z-stack is shown in each condition Scale bars: 10 µm. (G) Quantification of PLA signal density in each condition from two independent experiments (red and cyan). *P*-values (Mann–Whitney test) are shown. Sample sizes (embryos) in each replicate are as follows: no primary, *n*=6, 7; anti-NR5A2, *n*=8, 7; anti-KLF5, *n*=5, 7; anti-NR5A2+anti-KLF5, *n*=10, 8.

activation at the 8-cell stage. However, bulk CUT&Tag data cannot distinguish between whether a site is simultaneously bound by factors or whether the site is occupied by one factor in some cells and another factor in other cells. To test whether KLF5 and GATA6 can, in principle, co-bind to the same nucleosome with NR5A2, we purified recombinant full-length mouse NR5A2 as well as the

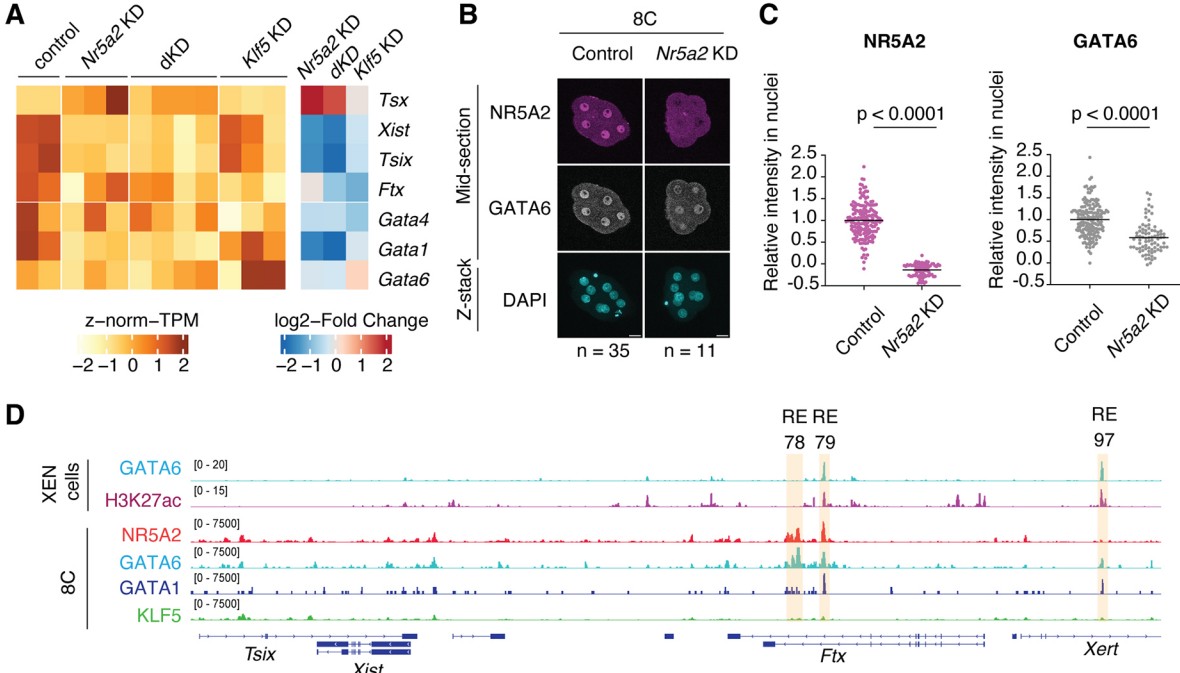

**Fig. 5. *Xist* expression is regulated by NR5A2 and GATA factors.** (A) RNA expression of XCI-related genes and GATA family members. Heatmaps show *z*-normalized TPM values for each replicate and log₂ fold-changes (*Nr5a2* KD/control, dKD/control and *Klf5* KD/control). (B,C) Representative images (B) and quantification (C) of immunostaining analysis showing NR5A2 (magenta), GATA6 (gray) and DAPI (cyan) in 8-cell embryos. NR5A2 and GATA signals are shown in mid-section images. DAPI signals are shown as full *z*-stack images. The number of embryos examined (*n*) from three independent experiments is indicated. Scale bars: 20 µm. Bars overlaid on the plots indicate means. *P*-values (*t*-test, two-sided) are shown. (D) IGV snapshot showing NR5A2 (red), GATA6 (light blue), GATA1 (dark blue) and KLF5 (green) near the *Xist* genomic locus. GATA6 and H3K27ac signals from extra-embryonic endoderm (XEN) cells are shown as control. Orange highlighted regions indicate *Xist* regulatory elements identified in previous studies (Gjaltema et al., 2022; Ravid Lustig et al., 2023).

DNA-binding domains (DBDs) of KLF5 and GATA6 (Fig. 6A). To test their DNA-binding specificities, we performed electrophoretic mobility shift analysis (EMSA). KLF5 DBD preferentially bound to the naked DNA containing its own motif, although some non-specific binding was observed (Fig. S9A). GATA6 DBD bound specifically to its own motif (Fig. S9A), demonstrating that the purified proteins selectively recognize their target DNA sequences.

To examine direct interactions between nucleosomes and NR5A2, KLF5 and GATA6 (hereafter referred to as NKG), we selected a nucleosome-enriched region measured by MNase-seq (GSE82127) in mouse ESCs (mESCs) and targeted by NKG at the 8-cell stage (Fig. 6B). *Nr5a2* KD and *Nr5a2 Klf5* dKD resulted in reduced H3K27ac, indicating that NR5A2 contributes to establishing active chromatin at this genomic locus. The selected DNA sequence of 149 bp is a part of *SINE B1/Alu*, which contains motifs for NR5A2 (5′-TCAAGGCCA-3′), KLF5 (5′-GGTGTGG-3′) and two separated GATA6 partial motifs (5′-GAT-3′) that can be recognized by one of the tandem zinc finger domains (Fig. 6C). We reconstituted the nucleosome-containing selected *SINE B1* DNA, resulting in NR5A2, KLF5 and GATA6 motifs located near superhelical locations +2.5, −4 and −2/+4, respectively. Single-particle analysis using cryogenic electron microscopy (cryo-EM) was performed in the presence of the single-chain variable fragment (ScFv) of a nucleosome antibody, which can stabilize the nucleosomes by binding to the acidic patch regions on histones (Zhou and Bai, 2021). We determined the nucleosome-containing selected mouse endogenous DNA at 4.1 Å resolution without crosslinking (Fig. 6D, Fig. S9B-E). The nucleosomal DNA was tightly wrapped around histones, showing that the nucleosome containing *SINE B1* DNA (hereafter referred to as B1 nucleosome) forms a canonical nucleosome structure.

We then tested NKG binding specificity to the B1 nucleosome. We detected specific band shifts corresponding to NR5A2-, KLF5 DBD- or GATA6 DBD-B1 nucleosome complexes, suggesting that the selected endogenous nucleosome can be targeted by these three TFs *in vitro* (Fig. 6E,F, Fig. S9F). Competition assays revealed that NR5A2- and GATA6 DBD-B1 nucleosome complexes were displaced by a 10-20× excess of specific competitor DNA, but not by the same amount of non-specific competitor DNA (Fig. S9G, lanes 1-6 and 17-22). In contrast, a 20× excess of non-specific competitor DNA largely outcompeted KLF5 DBD from the nucleosome (Fig. S9G, lane 16), while 2-10× excess of non-specific competitor DNA still allowed detectable KLF5 DBD-B1 nucleosome complex (Fig. S9G, lanes 7-15). Thus, NR5A2, KLF5 and GATA6 directly engage with their own motifs on B1 nucleosomal DNA.

To determine whether NR5A2 co-binds nucleosomes with either KLF5 or GATA6, KLF5 DBD or GATA DBD (0.75 µM) was incubated with the B1 nucleosome in the presence of a low amount of NR5A2 (0.25 µM). Higher-order complexes with slower migration than the NR5A2-B1 nucleosome complex were detected (Fig. 6G, lanes 2-4), suggesting that NR5A2 co-binds to the nucleosome with KLF5 or GATA6. Lastly, we tested whether NR5A2 co-binds with both KLF5 and GATA6 to the B1 nucleosome using a slightly reduced concentration of each TF to limit the total amount of protein. Higher-order complexes with slower migration than the NR5A2-KLF5 and NR5A2-GATA6 complexes were detected upon addition of all three TFs (Fig. 6H, lane 5, Fig. 6I). These *in vitro* binding assays suggest that the pioneer TFs NR5A2, KLF5 and GATA6 can simultaneously target nucleosomes containing their recognition motifs *in vitro*.

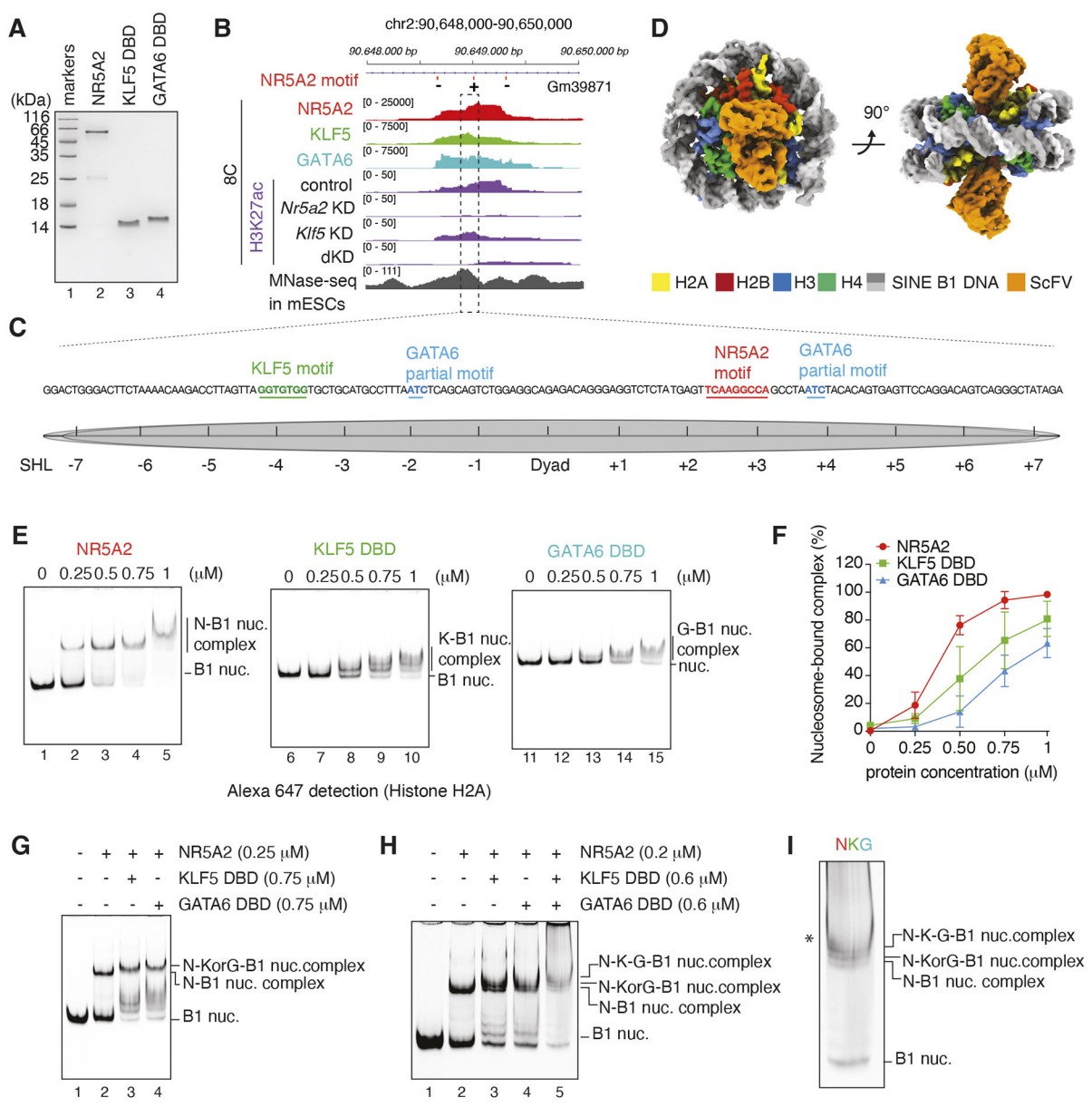

**Fig. 6. Nucleosome binding by pioneer TFs NR5A2, KLF5 and GATA6.** (A) Purified mouse NR5A2 full-length (lane 2), KLF5 DBD (lane 3) and GATA6 DBD (lane 4). Lane 1 shows molecular markers. (B) IGV snapshot showing co-occupancy by NR5A2, KLF5 and GATA6 along with MNase-seq profiles in mESCs. The dashed rectangle highlights the region used for mono-nucleosome reconstitution. 8C, 8-cell stage. (C) The sequence of a 149 bp DNA fragment containing *SINE B1/Alu* was used for mono-nucleosome reconstitution. NR5A2 (red), KLF5 (green) and GATA6 (blue) motifs are highlighted. SHL, superhelical location. (D) The cryo-EM structure of B1-nucleosome bound by ScFV. (E,F) Representative gel images (E) and quantification (F) of EMSA. Nucleosomes were analyzed by 5% native-PAGE and detected by Alexa Fluor 647 fluorescence. Data are shown as mean±s.d. for three independent experiments. (G) Representative EMSA gel showing TFs co-binding on the B1-nucleosome. Nucleosomes were analyzed by 6% native-PAGE and detected by Alexa Fluor 647 fluorescence. Reproducibility was confirmed with three independent experiments. (H) Representative EMSA gel showing NKG (NR5A2-KLF5-GATA6) co-binding on the B1-nucleosome. Nucleosomes were analyzed by 6% native-PAGE and detected by Alexa Fluor 647 fluorescence. Reproducibility was confirmed with three independent experiments. (I) Enlarged image of lane 5 from H. An asterisk marks an additional band, potentially non-specific KLF5 DBD binding to the NKG-B1 nucleosome complex.

## DISCUSSION
In this study, we show that NR5A2 chromatin binding changes with every cell division during the totipotency-to-pluripotency transition. As predicted from gene ontology analysis, NR5A2 regulates genes associated with the ICM, the TE and the Hippo signaling pathway. NR5A2 regulates the expression of KLF and GATA family TFs, which in turn function as co-regulators of NR5A2 during the totipotency-to-pluripotency transition. We provide evidence based on

CUT&Tag of embryos, PLA of individual cells in embryos and *in vitro* nucleosome binding assays that NR5A2 co-binds with lineage-determining TFs at certain genomic regions. Our data suggest that KLF5 contributes to H3K27ac deposition at genomic regions co-occupied by NR5A2, while GATA1 and GATA6 bind to distal *Xist* enhancers in conjunction with NR5A2. Thus, NR5A2 is a key factor that orchestrates gene regulatory networks involved in lineage-determining factors prior to cell fate commitments. Our findings

highlight how feed-forward regulatory loops by NR5A2 ensure robust gene activation during pre-implantation development (Fig. 7A).

Our genomics assays revealed that NR5A2 promotes chromatin accessibility and facilitates H3K27ac deposition as a pioneer TF. The exact mechanisms by which NR5A2 leads to chromatin reorganization and regulates epigenetic states remain unknown. Our cryo-EM analysis of the NR5A2-nucleosome complex revealed that NR5A2 partially unwraps nucleosomal DNA at the entry-exit site (Kobayashi et al., 2024). This structural change on the nucleosome is presumably required for local chromatin opening by evicting linker histone H1. Given that orphan nuclear receptors are associated with ATP-dependent chromatin remodelers and histone acetyltransferase p300 (Adachi et al., 2018; Chervova et al., 2024), NR5A2 may recruit these factors to disrupt the nucleosome and promote deposition of active chromatin marks at regulatory elements (Fig. 7B, i).

KLF5 is required for proper pre-implantation development and regulates lineage specification (Ema et al., 2008; Lin et al., 2010; Kinisu et al., 2021). In trophoblast stem cells, KLF5 maintains open chromatin and deposits H3K27 at TSSs through interaction with p300 (Dou et al., 2024). Despite its importance, KLF5 perturbation has relatively little effect in 8-cell embryos, possibly due to redundancy with KLF4 or other KLF family members (Kinisu et al., 2021). Intriguingly, our data indicate synergistic roles for NR5A2 and KLF5 in embryonic development (Fig. 3E,F) and H3K27ac deposition (Fig. 4C,D). One possible explanation is that KLF5 requires NR5A2 chromatin binding, thereby recruiting histone acetyltransferase to promote active chromatin. Since KLF5 expression is regulated by NR5A2, we could not directly test whether NR5A2 facilitates KLF5 chromatin binding by siRNA-mediated knockdown. An acute protein-depletion approach for short-term perturbation is required to dissect how these TFs regulate each other. We propose that NR5A2 functions together with KLF5 at a subset of regulatory elements to establish active chromatin during the totipotency-to-pluripotency transition (Fig. 7B, ii).

A notable finding is that NR5A2 binding is highly dynamic and undergoes changes with each blastomere division. In addition, NR5A2 targeting to *SINE B1/Alu* elements gradually decreases at later developmental stages. The mechanisms by which pioneer TFs selectively bind their own motifs in chromatin remain an open question. Previous studies have proposed that an increase in pioneer TF concentration correlates with chromatin opening and binding at new genomic regions (Blassberg et al., 2022; Gibson et al., 2024). Consistent with this, the number of NR5A2 peaks gradually increases concomitantly with higher RNA expression at the 8-cell stage, suggesting that local high concentration of NR5A2 may increase the chance of chromatin-binding events and keep chromatin open. Another possibility involves the regulation of *SINE B1/Alu* accessibility. The shutdown of *SINE B1/Alu* transcription may modulate the NR5A2 binding distribution. It has been reported that the KRAB (Krüppel-associated box) zinc finger protein ZFP266 binds to *SINE B1/Alu* and impedes chromatin opening by pluripotent factors (Kaemena et al., 2023). It is conceivable that embryo-specific KRAB zinc finger proteins play a specific role in repressing *SINE B1/Alu* elements during pre-implantation development.

Recently, a model has been proposed in which motifs on nucleosomes can act as signpost elements to direct TF combinatorial binding to enhancers (O'Dwyer et al., 2025). It would be interesting to determine the motif requirements, such as directionality and spacing, that govern the combinatorial binding of nuclear receptors, KLF, and GATA factors during development. Taken together, our study revealed the dynamics of chromatin binding profiles of TFs during the totipotency-to-pluripotency transition and provided fundamental insights into how TFs ensure robust gene activation through feed-forward loops.

## Study limitations

Since this study is based on siRNA-mediated KD approaches, we cannot exclude the possibility that there is some degree of inefficiency in the KD assay. KLF5 gross abundance is decreased

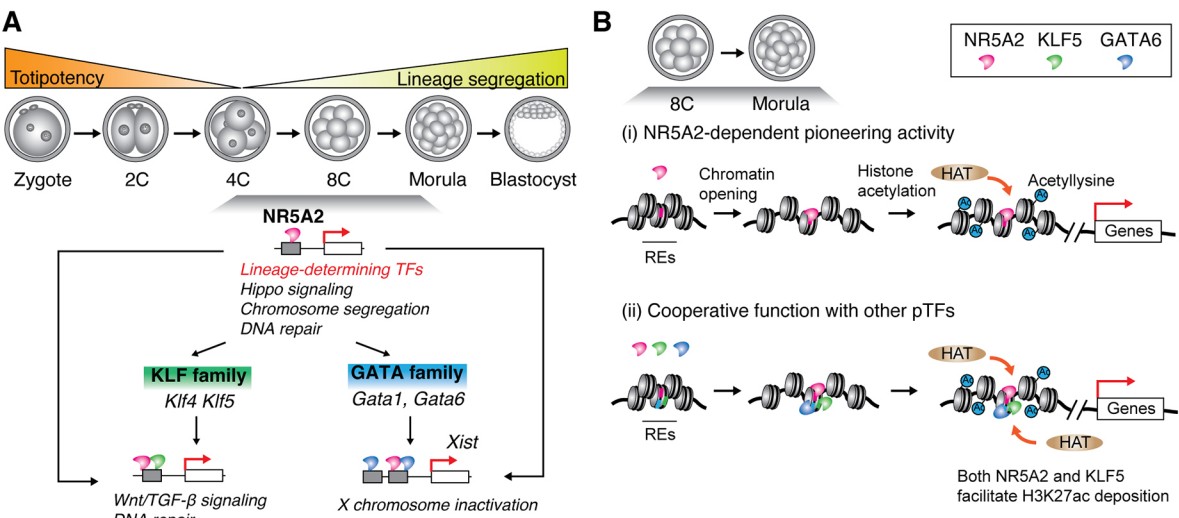

**Fig. 7. Feed-forward loop regulation by NR5A2 during the totipotency-to-pluripotency transition.** (A) Model of feed-forward loop mediated by NR5A2. Following zygotic genome activation, NR5A2 regulates a broad set of genes involved in lineage-determining factors, Hippo signaling, chromosome segregation and DNA repair. NR5A2 regulates the expression of KLF and GATA family TFs, which subsequently function as co-regulators of NR5A2. KLF5 and GATA6 co-occupy the chromatin with NR5A2 and further regulate transcription, such as Wnt signaling and *Xist*, ensuring proper embryonic development. (B) Proposed models of NR5A2 pioneer factor function in the mouse pre-implantation embryos. (i) NR5A2 recognizes and binds to its own motif sequence on nucleosomal DNA. NR5A2 locally opens the closed chromatin and recruits a histone acetyltransferase (HAT), such as p300, to establish active promoters or enhancers. (ii) Following local chromatin opening, both NR5A2 and KLF5 recruit a HAT to enhance histone acetylation for transcriptional activation. GATA6 also potentially binds to nucleosomes and stimulates transcription through its pioneering activity. 2C, 2-cell stage; 4C, 4-cell stage; 8C, 8-cell stage; REs, regulatory elements.

upon KD. However, there are precedents for chromatin-binding proteins that the chromatin-bound fraction can be difficult to deplete, e.g. auxin-mediated degradation of CTCF results in a gross loss of CTCF but not at some CTCF-bound sites (Luan et al., 2021). In this scenario, the *Nr5a2 Klf5* dKD would have a stronger effect on KLF5 depletion than *Klf5* KD because KLF5 synthesis is strongly reduced in the former, providing a potential explanation for the severe developmental phenotype of *Nr5a2 Klf5* dKD embryos.

## MATERIALS AND METHODS
### Animals
All animals housed at MPIB were sacrificed prior to the removal of organs in accordance with the European Commission Recommendations for the euthanasia of experimental animals (Part 1 and Part 2). Breeding and housing as well as the euthanasia of the animals were fully compliant with all German (e.g. German Animal Welfare Act) and EU (e.g. Directive 2010/63/EU) applicable laws and regulations concerning care and use of laboratory animals. Mice were kept at a daily cycle of 14-h light and 10-h dark with access to food *ad libitum*. Mice were bred in the MPIB animal facility.

### Collection of mouse embryos
We used B6CBAF1 frozen zygotes (Janvier Labs) for genome-wide mapping of NR5A2, KLF5, GATA6, H3K27ac and ATAC-seq. Zygotes were thawed according to a manual of kits of frozen zygotes and cultured in potassium-supplemented simplex-optimized medium (KSOM) with a lumox dish (Sarstedt). Embryos at the 2-cell, 4-cell, uncompacted 8-cell and morula (~32-cells without fluid-filled cavity) stages were collected after thawing at 22-24 h (34-36 hpf), 29-31 h (41-43 hpf), 44-46 h (56-58 hpf) and 59-60 h (71-72 hpf), respectively. For KD experiments followed by RNA-seq, H3K27ac CUT&Tag and ATAC-seq, zygotes were collected from superovulated B6129F1 female mice (3-5 weeks) mated with B6CBAF1 male mice (>8 weeks). To induce superovulation, females were injected with pregnant mare serum gonadotropin (5 IU) followed by human chorionic gonadotropin injection 46-48 h later. Zygotes were obtained by opening the oviduct 19 h after human chorionic gonadotropin injection.

### Immunofluorescence
The zona pellucida was removed by treatment with acidic Tyrode's solution. Embryos were fixed in 4% formaldehyde for 30 min and then permeabilized with 0.5% Triton X-100 in PBS. Blocking was carried out in 1% bovine serum albumin (BSA; Sigma-Aldrich, A8577) in PBTX (0.1% Triton X-100 in PBS) for 1 h at room temperature. Cells were incubated with primary antibodies (NR5A2: 1:100, R&D Systems, PP-H2325-00; KLF5: 1:100, Proteintech, 21017-1-AP; GATA6: 1:500, R&D Systems, AF1700; CDX2: 1:200, BioGenex, MU392A-UC; NANOG: 1:500, Active Motif, 61419) for 1 h at room temperature on a nutator. Cells were washed through blocking solution three times with PBTX and were incubated with the appropriate secondary antibody (1:500, Thermo Fisher Scientific, goat anti-mouse Alexa Fluor 488 or 647, goat anti-rabbit Alexa Fluor 568, donkey anti-goat Alexa Fluor 594) for 30 min at room temperature. After three washes with PBTX, cells were mounted onto SUPERFROST Plus Adhesion microscope slides with VECTASHIELD with DAPI. Images were acquired using a confocal microscope (Leica, STELLARIS 5). All images were analyzed with Fiji and Arivis Pro/Vision4D. The signal intensity within nuclei of control samples was determined, and the cytoplasmic signal was subtracted as background. The average signal intensity of the control samples was set as 1.0.

### Microinjection of zygotes for siRNA knockdown
For siRNA knockdown, isolated zygotes (8-9 hpf) were microinjected with siRNAs against targets *Nr5a2* (5 µM), *Klf5* (10 µM) or control (15 µM). As a control of successful injection, 150 ng/µl H2B-EGFP was co-injected. The siRNA targeting control and *Nr5a2* (Lai et al., 2023) were ordered as *Silencer* Select from Life Technologies. A predesigned siRNA (160900, Silencer™ pre-designed siRNA, Life Technologies) was used for Klf5 KD. To test *Gata6* KD efficiency, 100 µM of siRNA targeting *Gata6* (#158651, Silencer pre-designed siRNA, Life Technologies) was used. The injected

embryos were cultured in KSOM using lumox dishes (35 mm, Sarstedt) under 5% $O_2$, 5% $CO_2$, 90% $N_2$.

### Embryonic development assay
Embryos were cultured in 30 µl microdrops of KSOM using lumox dishes (35 mm, Sarstedt) under 5% $O_2$, 5% $CO_2$, 90% $N_2$. Embryonic development was monitored once every 24 h using a stereomicroscope with a camera (Flexacam C3, Leica Microsystems).

### PLA
PLA was performed using Duolink assay reagents (Sigma-Aldrich). Zona pellucida was removed by treatment with acidic Tyrode's solution. Embryos were fixed in 4% formaldehyde for 30 min and then permeabilized with 0.5% Triton X-100 in PBS. Blocking was carried out in 1% BSA (Sigma-Aldrich, A8577) in PBTX (0.1% Triton X-100 in PBS) for 1 h at 37°C. Cells were incubated with primary antibodies (NR5A2: 1:100, R&D Systems, PP-H2325-00; KLF5: 1:100, Proteintech, 21017-1-AP) in blocking solution for 1 h at room temperature on a nutator. Embryos were washed through 1× Duolink washing buffer A three times and were incubated with probe mixture (rabbit PLUS probe DUO92002 and mouse MINUS probe DUO92004) for 1 h at 37°C. After three washes with 1× Duolink washing buffer A, embryos were incubated with the ligation mixture for 30 min at 37°C. After three washes with 1× Duolink washing buffer A, embryos were incubated with the amplification mixture for 100 min at 37°C. Embryos were washed through 1× Duolink washing buffer B twice, followed by a quick wash in PBS. Embryos were incubated with VECTASHIELD with DAPI and mounted. Images were acquired using a confocal microscope (Leica, STELLARIS 5). All images were analyzed with custom Fiji and R scripts. Only PLA spots located within each nucleus were counted and used for further analysis. The PLA density was calculated as total number of PLA spots across all image stacks per total area of nucleus. The Mann–Whitney test was used for statistical comparison of PLA density between two conditions.

### CUT&Tag
CUT&Tag was performed as described previously (Gassler et al., 2022). Embryos with intact zona pellucida were incubated with ice-cold extraction buffer (25 mM HEPES-NaOH, pH 7.4, 50 mM NaCl, 3 mM $MgCl_2$, 300 mM sucrose and 0.5% Triton X-100) on ice for 6-7 min and were washed three times with ice-cold extraction buffer without Triton X-100. Pre-extracted embryos were immediately lightly fixed in Dulbecco's phosphate-buffered saline (DPBS) with 0.1% formaldehyde for 2 min at room temperature. Embryos were incubated in antibody buffer [20 mM HEPES-NaOH, pH 7.5, 150 mM NaCl, 0.5 mM Spermidine, 0.1% BSA, 2 mM EDTA and 1× Protease inhibitor cocktail (Roche) with primary antibody [1:100; NR5A2 (R&D Systems, PP-H2325-00), KLF5 (Proteintech, 21017-1-AP), GATA6 (R&D Systems, AF1700), H3K27ac (Active motif, 39133)] overnight at 4°C on nutator. Embryos were washed three times through antibody buffer and further incubated with secondary antibody [1:100, guinea pig anti-rabbit H&L antibody (ABIN101961, Antibodies-Online) for anti-rabbit primary antibody, rabbit anti-mouse IgG H&L (ab46540, abcam) for anti-mouse primary antibody, and rabbit anti-goat IgG H&L (ab6697, abcam) for anti-goat primary antibody] for 1.5 h at room temperature. Cells were washed three times with Wash buffer [20 mM HEPES-NaOH, pH 7.5, 150 mM NaCl, 0.5 mM Spermidine and 1× Protease inhibitor cocktail (Roche)] and further incubated with homemade pA-Tn5 or pAG-Tn5 adaptor complex (1:250) for 1.5 h at room temperature. Embryos were washed three times through Wash-300 buffer [20 mM HEPES-NaOH, pH 7.5, 300 mM NaCl, 0.5 mM Spermidine and 1× Protease inhibitor cocktail (Roche)]. Embryos were transferred into a 1.5 ml DNA LoBind Tube (Eppendorf) with 200 µl Tagmentation buffer (10 mM $MgCl_2$ in Wash-300) and incubated at 37°C for 1 h. The samples were then deproteinized and reverse-crosslinked. The DNA was purified by phenol-chloroform extraction and followed by ethanol precipitation. The DNA was amplified in a 50 µl reaction with 1× Q5 High-Fidelity Master mix (NEB) and 0.25 µM barcoded primers using the following PCR program: 72°C for 5 min; 98°C for 30 s; 15 cycles of 98°C for 10 s and 63°C for 10 s; final extension at 72°C for 1 min and hold at 4°C. After the PCR reaction, libraries were purified with 1.1× AMPure XP magnetic beads (Beckman Coulter) and were eluted in 20 µl RNase-free water. Purified DNA libraries were sequenced

on a NextSeq 500 (Illumina), Novaseq 6000 (Illumina) or AVITI (Element Biosciences) with paired-end reads.

## TIP-seq

TIP-seq was performed according to the original protocol (Bartlett et al., 2021). The procedure was carried out in the same manner as the CUT&Tag method until the Tn5-mediated DNA tagmentation step. DNA tagmentation reaction was conducted in 50 µl Tagmentation buffer using Axygen 0.2 ml MAXYMum Recovery Thin Wall PCR Tubes. The samples were then deproteinized and reverse-crosslinked by adding 1.7 µl 0.5 M EDTA, 1 µl 10% SDS (0.2% final) and 0.5 µl 20 mg/ml proteinase K (Roche, 03115887001) and incubating for 2 h at 50°C, followed by overnight incubation at 65°C. After adding 2 µl of homemade spike-in *Drosophila* DNA, tagmented gDNA was purified using 2× volume of AMPure XP magnetic beads (Beckman Coulter), and DNA and beads were resuspended in 8 µl RNase-free water. The further steps involving gap filling, *in vitro* transcription, cDNA synthesis and cDNA tagmentation were performed according to the original TIP-seq protocol (Bartlett et al., 2021). The tagmented DNA was amplified in a 50 µl reaction with 1×Q5 High-Fidelity Master mix (NEB) and 0.25 µM barcoded primers using the following PCR program: 72°C for 5 min; 98°C for 30 s; 12 cycles of 98°C for 10 s and 63°C for 10 s; final extension at 72°C for 1 min and hold at 4°C. After the PCR reaction, libraries were purified by 1.1× AMPure XP magnetic beads (Beckman Coulter) and were eluted in 20 µl RNase-free water.

## CUT&RUN

CUT&RUN was performed according to the published protocol (Hayashi and Inoue, 2023) with a few modifications. Eight-cell embryos were treated with acidic Tyrode's solution to remove the zona pellucida, and washed through 0.15% BSA in PBS on a microplate (Nunc™ Microwell™ Minitrays). Embryos were incubated in antibody buffer [20 mM HEPES-NaOH, pH 7.5, 150 mM NaCl, 0.5 mM Spermidine, 0.02% Digitonin, 2 mM EDTA and 1× Protease inhibitor cocktail (Roche) with primary antibody (1:100, GATA1 (abcam, ab11852)] overnight at 4°C on a nutator. Embryos were washed three times through antibody buffer and further incubated with secondary antibody (1:100, guinea pig anti-rabbit H&L antibody (ABIN101961, Antibodies-online) for 1 h at room temperature. After incubation, embryos were washed three times through cold Dig-Wash buffer (20 mM HEPES-NaOH, pH 7.5, 150 mM NaCl, 0.5 mM Spermidine, 0.02% digitonin and 1× Protease inhibitor cocktail), followed by incubation in Dig-Wash buffer for 20 min at room temperature. Embryos were further incubated with cold Dig-Wash buffer containing homemade pAG-MNase (500 ng/µl) for 1.5 h on ice with gentle nutation. Embryos were washed twice in cold Dig-Wash buffer and incubated for 20 min at room temperature with gentle nutation. Embryos were transferred into a 1.5 ml tube containing 50 µl of the wash buffer and then captured by Bio-Mag Plus Concanavalin A-coated beads. After removing the supernatant, 180 µl cold Dig-wash buffer was added, and MNase reaction was carried out in the presence of 2 mM CaCl₂ for 20 min on ice. The reaction was stopped by adding 20 µl 10XSTOP buffer (1700 mM NaCl, 500 mM EDTA, 22 mM EGTA, 0.02% digitonin, 250 µg/ml RNase A, 500 µg/ml glycogen). The protein–DNA complexes were then released by 20 min incubation at 37°C with shaking at 1000 rpm. The supernatant was transferred into a new 1.5 ml of DNA LoBind Tube (Eppendorf) and then deproteinized. The DNA was purified by phenol-chloroform extraction and followed by ethanol precipitation. DNA libraries were prepared using a NEBNext Ultra II DNA library preparation kit for Illumina (New England Biolabs) according to the manufacturer's instructions

## Omni ATAC-seq

Omni ATAC-seq was performed as described previously (Gassler et al., 2022). The zona pellucida was removed by treatment with acidic Tyrode's solution. Embryos were transferred into 1.5 ml of DNA LoBind Tube (Eppendorf) with 200 µl of PBS and were spun at 500 *g* at 4°C for 5 min to remove supernatant, then 50 µl of ATAC-Resuspension buffer (RSB) (10 mM Tris-HCl, pH 7.4, 10 mM NaCl, 3 mM MgCl₂) containing 0.1% NP-40, 0.1% Tween 20 and 0.01% digitonin was added to the tube and incubated for 3 min on ice. After incubation, 200 µl of ATAC-RSB containing 0.1% Tween 20 was further added to the tube to wash out the lysis. Nuclei were spun at 500 *g* at 4°C for 10 min to remove supernatant.

Nuclei were incubated with 50 µl of transposition mixture (10 mM Tris-HCl, pH 7.6, 5 mM MgCl₂, 10% dimethyl formamide, 33% PBS, 0.1% Tween 20, 0.01% digitonin and 95 nM Tn5-adaptor complex) at 37°C for 30 min with 1000 rpm mixing. The samples were then deproteinized and further purified by phenol-chloroform extraction and followed by ethanol precipitation. The DNA was amplified in a 50 µl reaction with 1×NEBNext HF 2×PCR Master mix (NEB) and 0.25 µM barcoded primers using the following PCR program: 72°C for 5 min; 98°C for 30 s; 15 cycles of 98°C for 10 s, 63°C for 30 s and 72°C for 1 min; final extension at 72°C for 5 min and hold at 4°C. After the PCR reaction, libraries were purified by 1.1× AMPure XP magnetic beads (Beckman Coulter) and were eluted in 20 µl of RNase-free water. Purified DNA libraries were sequenced on a NextSeq 500 (Illumina) or AVITI (Element Biosciences) with paired-end reads.

## RNA-seq library preparation

The zona pellucida was removed by treatment with acidic Tyrode's solution. Ten 8-cell embryos were washed five times in M2 medium and then quickly washed in PBS. Embryos were lysed in lysis buffer containing RNase inhibitor and ERCC spike-in RNA on ice for 15 min and then snap-frozen with liquid nitrogen. cDNA was prepared using a SMART-seq v4 Ultra-Low-Input RNA kit for Sequencing (Takara Bio, 634888). Library was prepared by a Nextera XT DNA Library Prep Kit (Illumina, FC-131-1024) according to the manufacturer's protocol. Purified DNA libraries were sequenced on a Novaseq 6000 (Illumina) with paired-end reads.

## Protein purification

His₆-tagged full-length mouse NR5A2 was purified as described previously (Gassler et al., 2022). The DNA fragment encoding *Mus musculus* KLF5 DBD (348-446) was ligated into BamHI-NotI sites of pGEX6P-1 vector. Glutathione-S-transferase (GST)-fused KLF5 DBD was expressed in *Escherichia coli* BL21 (DE3) codon plus RIL (Agilent Technologies). The cells were cultivated at 30°C, and the protein expression was induced with 0.25 mM IPTG at 16°C for 16-18 h. The cells were resuspended in buffer 1 (50 mM Tris-HCl, pH 7.5, 500 mM NaCl, 1 mM EDTA, 10% glycerol and 2 mM 2-mercaptoethanol). The cells were disrupted by sonication, and the cell debris was further removed by centrifugation at 39,191 *g* for 20 min. The protein bound to Glutathione Sepharose 4B beads (Cytiva) was washed by 50 column volumes with buffer 1. GST-tag was cleaved with homemade PreScission protease on the column at 4°C overnight. The supernatant containing KLF5 DBD was collected and concentrated with an Amicon Ultra centrifugal filter unit (Millipore).

*Mus musculus* GATA6 DBD was purified as described previously with a few modifications (Takaku et al., 2016). The codon-optimized DNA fragment encoding *Mus musculus* GATA6 DBD (382-492) was ligated into NdeI-BamHI sites of pET-15b vector with a N-terminal His₆-TEV cleavage site. His₆-tag GATA6 DBD was purified by Ni-NTA agarose beads (QIAGEN). His₆-tag was removed by homemade TEV protease, and GATA6 DBD was purified by MonoS 5/50 GL (Cytiva) column chromatography. The sample was further loaded onto Superdex200 16/60 (Cytiva), and fractions containing GATA6 DBD were stored at −70°C.

Mouse histones H2A K119C, H2B, H3.3 and H4 were expressed and purified as previously described (Kujirai et al., 2018). Histone H2AK119C-H2B complex and H3.3-H4 complex were reconstituted as previously described (Kobayashi et al., 2024). For the fluorescent labeling of H2A-H2B complex, 50 µM of H2A K119C-H2B complex were incubated with 500 µM Alexa Fluor 647 C₂ Maleimide (Invitrogen) in 20 mM Tris-HCl (pH 7.5) buffer containing 0.1 M NaCl and 1 mM TCEP at room temperature for 2 h in the dark. The reaction was stopped by the addition of 150 mM 2-mercaptoethanol, and the sample was then dialyzed against 20 mM Tris-HCl buffer containing 0.1 M NaCl, 1 mM EDTA and 5 mM 2-mercaptoethanol.

The DNA fragment encoding the ScFv linker 20 (ScFv²⁰) was synthesized by the PCR method. The amplified DNA fragment encoding ScFv²⁰ was ligated into the NdeI-BamHI site in the pET15b-TEV vector. ScFv²⁰ was expressed and purified as previously described (Zhou and Bai, 2021).

## DNA preparation

The endogenous DNA fragment containing SINE B1/Alu for nucleosome reconstitution were amplified by PCR and further purified by polyacrylamide

gel (6%) electrophoresis using a Prep Cell apparatus (Bio-Rad). The eluted DNA was concentrated with an Amicon Ultra centrifugal filter unit (Millipore).

## Preparation of the nucleosomes containing mouse endogenous DNA

Nucleosomes were reconstituted by the salt dialysis method. Briefly, the DNA, Alexa Fluor 647-labeled histone H2A-H2B complex, and H3.3-H4 complex were mixed in a 1:4:3.6 molar ratio in 2 M KCl high salt buffer. After salt dialysis, the reconstituted nucleosomes were further purified by polyacrylamide gel (6%) electrophoresis using a Prep Cell apparatus (Bio-Rad). The nucleosomes were concentrated with an Amicon Ultra centrifugal filter unit (Millipore).

## EMSA

For EMSA with short oligo DNA (Chen et al., 2012; Gassler et al., 2022), 10% non-denaturing polyacrylamide gels (0.5× TBE) were pre-run at 100 V for 30 min at 4°C. DNA (50 nM) was incubated with 0-1 µM KLF5 DBD or GATA6 DBD at room temperature for 30 min in reaction buffer (20 mM Tris-HCl, pH 7.5, 120 mM NaCl, 1 mM MgCl$_2$, 10 µM ZnCl$_2$, 1 mM DTT, 100 µg/ml BSA). After incubation, the samples were loaded onto gels, and electrophoresis was performed at 100 V for 100 min at 4°C. Gels were stained by SYBR Gold (Invitrogen) and were imaged using the ChemiDoc MP imaging system (Bio-Rad).

For EMSA with nucleosome, 5-6% non-denaturing polyacrylamide gels (0.5× TBE) were pre-run at 100 V for 30 min at 4°C. Nucleosomes (50 nM) were incubated with 0-1 µM NR5A2, KLF5 DBD or GATA6 DBD at room temperature for 30 min in reaction buffer (20 mM Tris-HCl, pH 7.5, 120 mM NaCl, 1 mM MgCl$_2$, 10 µM ZnCl$_2$, 1 mM DTT, 100 µg/ml BSA). For the competition assay, non-labeled naked DNA containing non-specific or specific sequence (0.1-1 µM) was added with the nucleosome. After incubation, the samples were loaded onto gels and electrophoresis was performed at 100 V for 80-120 min at 4°C. The gels were imaged by detecting Alexa Fluor 647 fluorescence and SYBR Gold (Invitrogen) using the ChemiDoc MP imaging system (Bio-Rad).

## Cryo-EM specimen preparation and data acquisition

ScFV[20] (1.5 µM) and the B1-nucleosome (0.5 µM) were mixed in a 3:1 molar ratio on ice for 30 min. The ScFV[20]–nucleosome complex was concentrated by Amicon Ultra centrifugal filter unit (Millipore) until 215 ng/µl (DNA). To prepare the cryo-EM specimen, the sample (4 µl) was applied to a glow-discharged holey carbon grid (Quantifoil R1.2/1.3 200-mesh Cu). The grids were blotted for 3.0 s at a blotting strength setting of 5 under 100% humidity at 4°C and then plunged into liquid ethane and cooled by liquid nitrogen using a Vitrobot Mark IV (Thermo Fisher). Data acquisition for the ScFV[20]–B1 nucleosome complex was conducted using a 200 kV Glacios (Thermo Fisher Scientific) equipped with a GIF Quantum 967 energy filter and a K3 direct electron detector (Gatan) running in correlated double sampling mode at a magnification factor of 36 kx, equivalent to a pixel size of 1.136 Å per pixel. Automated data acquisition was performed using SerialEM software. A total of 2489 videos were collected, with a total electron dose of 60 e/Å² fractionated over 40 frames.

## Image processing

Data processing was performed by RELION 4.0 (Kimanius et al., 2021). In total, 2489 micrographs were stacked and motion-corrected using MOTIONCOR2 (Zheng et al., 2017) with dose weighting. The estimation of contrast transfer function (CTF) from the dose-weighted micrographs was performed using CTFFIND4 (Rohou and Grigorieff, 2015). From 2449 micrographs, 571,540 particles were automatically picked and extracted with a binning factor of 4 (pixel size of 4.724 Å/pixel). In total, 516,292 particles were further selected by 2D classification. The *ab initio* model generated was used as the initial model for 3D classification. After two rounds of 3D classification, 152,279 particles were selected and re-extracted without binning for 3D refinement, followed by Bayesian polishing and CTF refinement. The refined map of the ScFV[20]–B1 nucleosome complex was sharpened with a B-factor of −120 Å². The resolution of the final 3D map was 4.09 Å, as estimated by the gold standard Fourier Shell Correlation (FSC) at FSC=0.143 (Rosenthal and Henderson, 2003). This map was subsequently post-processed through

local B-factor correction by DeepEMhancer (Sanchez-Garcia et al., 2021). The local resolution of the map was calculated by RELION4 and visualized with UCSF ChimeraX (version 1.5) (Pettersen et al., 2021).

## Data analysis

### CUT&Tag, ChIP-seq and ATAC-seq data processing

Low-quality reads were trimmed out using TrimGalore (version 0.6.2) (https://github.com/FelixKrueger/TrimGalore) using parameters –paired –quality 20 –length 20. Read mapping was done using Bowtie2 (version 2.3.5.1) (Langmead and Salzberg, 2012) using parameters -t -q -N 1 -L 25 -X 2000 –no-mixed –no-discordant. Unmapped reads and reads with low mapping quality (Q<30) were removed. PCR duplicated reads were identified and discarded using the 'MarkDuplicates' command from Picard (version 2.18.27) (http://broadinstitute.github.io/picard/) with parameters VALIDATION_STRINGENCY=LENIENT. The RPKM values were calculated to represent read coverage using the bamCoverage function from deepTools (version 3.5.4) using parameters –binSize 1 –ignoreForNormalization chrM (Ramirez et al., 2016). The RPKM values were further transformed using Z-score normalization for the visualization using custom R script. To access similarity and reproducibility of the libraries, correlation between libraries were calculated using the 'multiBigwigSummary' command from deepTools with parameters –chromosomesToSkip chrM, then heatmaps of library correlation were made using the 'plotHeatmap' command from deepTools with parameters –skipZeros –removeOutliers –corMethod pearson. Heatmaps of chromatin binding were generated using the 'computeMatrix' and 'plotHeatmap' commands from deepTools. The 'computeMatrix' command was run with the following parameters: –binSize 50 –missingDataAsZero –skipZeros.

Reads from Omni-ATAC-seq datasets were mapped to both mouse (mm10) and *Drosophila* (dm6, spike-in) genomes. Unmapped, low mapping quality, and PCR duplicated reads were removed as described previously. Note that normalization by spike-in reads was not performed due to extremely low mappable spike-in reads in these samples.

For analysis of ATAC-seq and H3K27ac CUT&Tag of *Nr5a2* and *Klf5* knockdown samples, the log2 fold change of chromatin coverage signals between the siRNA knockdown and control condition were calculated using the 'bigwigCompare' command from deepTools using the following parameters: –binSize 100 –skipZeroOverZero –operation log2.

### Differential binding analysis

The R package DiffBind (Ross-Innes et al., 2012; https://rdrr.io/bioc/DiffBind/f/inst/doc/DiffBind.pdf) were used to generate consensus peaks, to count reads, and to normalize the chromatin occupancy datasets. Normalized read counts within consensus peaks were used to build the principal component analysis. The dba.analyze function was used for differential binding analysis of datasets with at least three replicates per condition, such as H3K27ac datasets. For datasets that did not meet this requirement, occupancy information, including consensus peak, raw read counts within peaks, and normalizing factors, were extracted from DiffBind's DBA class object. The extracted information was then used to build the DESeq2 DESeqDataSet object and perform differential binding analysis using DESeq function.

### Annotation of chromatin states

To infer chromatin state based on epigenetic status, ATAC-seq and ChIP-seq datasets of H3K4me3, H3K27ac, H3K27me3 and H3K9me3 from 2-cell embryos to mESCs were collected as specified in Table S3 (Dahl et al., 2016; Liu et al., 2016; Zhang et al., 2016; Wang et al., 2018). ChromHMM (version 1.23) (Ernst and Kellis, 2012, 2017) was used to categorize chromatin states based on these signals. The data were first binarized using the 'BinarizeBam' command with the default parameters. Then the 'LearnModel' command was used to infer 3-15 chromatin states based on the same input data. The results with 11 states were arbitrarily chosen for further analysis. The assignment of chromatin state to active (promoter or enhancer), repressive or open chromatin states was based on emission signal as shown in Fig. S1D.

### RNA-seq data processing

Adaptor sequences were trimmed from the reads using TrimGalore (version 0.6.2) (https://github.com/FelixKrueger/TrimGalore) using parameters

–paired –quality 10 –length 10. Reads were then mapped to the custom reference transcriptome based on GRCm30 (version 102) with addition of H2B-EGFP and ERCC spike-in reference using STAR with default setting (Dobin et al., 2013). Expression of individual genes was quantified using featureCounts (version 2.0.1) (Liao et al., 2014) with the additional parameters -t exon -g gene_id -p -O –fraction. To identify differentially expressed genes (DEGs), expression data were imported to R. Genes with low counts across all samples were identified and removed using the HTSFilter package (Rau et al., 2013). DEGs were identified using the DESeq2 package (Love et al., 2014). Only genes with $P$-value cut-off of 0.05 and with at least a 2-fold change compared with the control condition were further considered as DEGs. Gene ontology analysis of DEGs and genes associated with NR5A2 peak clusters was performed using the clusterProfiler package in R.

## Peak calling and motif analysis
Peak calling for CUT&Tag, ATAC-seq and ChIP-seq datasets was carried out using the MACS2 'callpeak' function (version 2.2.9) with default parameters (Zhang et al., 2008). The identification of *de novo* motif analysis for TF CUT&Tag peaks was achieved using the 'findMotifsGenome' command from HOMER (version 4.10) (Heinz et al., 2010) with the '-size given' parameter for most of the datasets, and with '-size 500' parameter for cluster analysis of *Nr5a2* peaks in the morula stage. The peak lists are shown in Table S1.

## NR5A2 binding site clustering and association with gene expression
A custom R script was used to identify all NR5A2 binding loci from 2-cell to morula stages. NR5A2 binding signal (in RPKM) at these loci were collected using the deepTools 'computeMatrix' command with the following options: –binSize 50 –missingDataAsZero –skipZeros. The output matrix was used as an input in the deepTools 'plotHeatmap' command for heatmap visualization and clustering of the binding using –kmeans 6, and using –outFileSortRegions option to retrieve the clustering of NR5A2 peaks. For the analysis of gene expression in Fig. 1F, a gene was associated with the NR5A2 peak cluster nearest to its TSS if the following criteria were met: (1) the nearest NR5A2 peak was located within 10 kb of its TSS, and (2) the gene had expression of more than 5 transcripts per million (TPM) in at least in one stage of preimplantation development.

## Acknowledgements
We thank R. Hornberger and C. Kobayashi for technical assistance. We thank D. Bollschweiler and T. Schäfer at the cryo-EM facility (RRID:SCR_025744) for assistance in cryo-EM data collection, R. H. Kim, A. Casper and R. Gautsch for sequencing at the NGS facility (RRID:SCR_025746), and the animal facility. K.T. is an Honorary Professor at the Department of Biology, Ludwig-Maximilians-University (LMU), Munich, Germany. S.R. and A.M. are IMPRS PhD students affiliated with the Department of Biology, LMU.

## Competing interests
The authors declare no competing or financial interests.

## Author contributions
Conceptualization: W.K., S.R., K.T.; Data curation: W.K., S.R., E.N.A., A.M., K.T.; Formal analysis: W.K., S.R.; Funding acquisition: K.T.; Investigation: W.K., E.N.A., A.M.; Methodology: W.K.; Project administration: K.T.; Resources: K.T.; Supervision: K.T.; Validation: W.K., S.R., A.M.; Visualization: W.K., S.R., E.N.A., A.M.; Writing – original draft: W.K., S.R., K.T.; Writing – review & editing: W.K., S.R., K.T.

## Funding
European Research Council grant ERC-CoG-818556 TotipotentZygotChrom (K.T.), Max Planck Society (K.T.). Open Access funding provided by the Max Planck Society. Deposited in PMC for immediate release.

## Data and resource availability
The raw sequencing data and processed data for this study have been deposited in the Gene Expression Omnibus (GEO) database under accession number GSE289580. The custom code for PLA can be found at https://github.com/TotipotencyLab/Kobayashi_etal_Development_2025. All other relevant data and details of resources can be found within the article and its supplementary information.

## Peer review history
The peer review history is available online at https://journals.biologists.com/dev/lookup/doi/10.1242/dev.205059.reviewer-comments.pdf

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
