## [Peer Review File · Development (Cambridge, England)]

Feed-forward loops by NR5A2 ensure robust gene activation during pre-implantation development

Wataru Kobayashi, Siwat Ruangroengkulrith, Eda Nur Arslantas, Adarsh Mohanan and Kikue Tachibana

DOI: 10.1242/dev.205059

Editor: Peter Rugg-Gunn

Review timeline

Original submission:	26 June 2025
Editorial decision:	5 August 2025
First revision received:	3 November 2025
Accepted:	24 November 2025

Original submission

First decision letter

MS ID#: dev.205059

MS TITLE: Feed-forward loops by NR5A2 ensure robust gene activation during pre-implantation development

AUTHORS: Wataru Kobayashi; Siwat Ruangroengkulrith; Eda Nur Arslantas; Adarsh Mohanan; Kikue Tachibana

Dear Dr Tachibana,

I have now received all the referees' reports on the above manuscript, and have reached a decision. The referees' comments are appended below, or you can access them online: please go to: *****

As you will see, the referees express interest in your work, but have some significant criticisms and recommend a substantial revision of your manuscript before we can consider publication. If you are able to revise the manuscript along the lines suggested, which may involve further experiments, I will be happy receive a revised version of the manuscript. Your revised paper will be re-reviewed by one or more of the original referees, and acceptance of your manuscript will depend on your addressing satisfactorily the reviewers' major concerns. Please also note that Development will normally permit only one round of major revision. If it would be helpful, you are welcome to contact us to discuss your revision in greater detail.

Please attend to all of the reviewers' comments and ensure that you clearly highlight all changes made in the revised manuscript. Please avoid using 'Tracked changes' in Word files as these are lost in PDF conversion. I should be grateful if you would also provide a point-by-point response detailing how you have dealt with the points raised by the reviewers in the 'Response to Reviewers' box. If you do not agree with any of their criticisms or suggestions please explain clearly why this is so.

Reviewer 1

SUMMARY OF THE ADVANCE MADE IN THIS PAPER AND ITS POTENTIAL SIGNIFICANCE TO THE FIELD

In this manuscript Kobayashi et al investigate how the pioneer TF Nr5a2 functions during early mouse development, an interesting question for the field of developmental biology. They characterize genome-wide binding profiles for Nr5a2 at different stages of preimplantation development and identify KLF and GATA factor motifs enriched withing Nr5a2 peaks, suggesting that these factors co-regulate active enhancers with Nr5a2. They follow up with functional experiments on the roles of Nr5a2, Klf5 and their combined effect on preimplantation development, revealing that the two factors function synergistically in blastocyst formation. They also use an in vitro assay to show that Nr5a2, Klf5 and Gata6 can bind to the same nucleosome, supporting the model that these factors can act together at the same site. There is a wealth of experimental data in this manuscript, but the story is somewhat incoherent in places.

SUGGESTIONS TO AUTHORS

Major:

-The claim that Nr5a2 regulates Klf5 and Gata6 is not supported.

Klf5 transcript and protein are indeed down upon Nr5a2 KD, but whether Nr5a2 binds in the vicinity of Klf5 and whether this binding site changes (accessibility or H3K27ac) upon Nr5a2 KD is not examined.

The direct regulation of Gata6 by Nr5a2 is not evident. Gata6 transcripts are negligibly downregulated in Nr5a2 KD and it is difficult to comprehend how the mild down-regulation of Gata6 protein is due to Nr5a2 KD. Would Nr5a2 regulate Gata6 post-transcriptionally? Also, Figure 5D shows Nr5a2 binding at a site near Gata6, but in the KD chromatin accessibility and H3K27ac do not change at this site.

In sum, regulation of Klf5 needs more evidence and regulation of Gata6 is not supported by the data.

- Along the same lines, this raises the concern that the proposed Nr5a2-Gata6 link for X inactivation are not supported by the data. The down-regulation of Gata1 transcript and the potentially direct role of Nr5a2 in this better supported. However, the authors do not examine the binding of Gata1 in their study.

-The section on "KLF5 directly regulates other KLF family members" seems like an afterthought and does not add much to the main message of the paper. It is also unclear how this function of Klf5 relates to Nr5a2.

-The analysis of phenotypes in KDs (Figure 3) is quite rudimentary. It is sufficient to make the claim of cooperative action between Nr5a2 and Klf5, however, it would be a lot more informative if the authors could perform lineage-specific analysis, including immunofluorescent stainings for cell fate markers and cell counts for each cell type. This could be particularly interesting as the authors later show that genes involved in the first cell fate decision are affected in KDs.

Minor:

-It is somewhat unclear to me if Nr5a2 is only acting at repetitive elements or also at other genomic sites?

-Please specify what morula is. Is it referring to compacted 8 and 16-cell stage embryos? Whereas "8-cell stage" is referring to uncompact 8-cell embryos?

-Consider moving Figure S1C to main figure, as this is an important overview of Nr5a2 binding dynamics across embryo stages.

-Line 116 "until cleavage-stage blastomeres" - as far as we know cleavage divisions occur until the blastocyst stage, so it would be preferred to use a more concrete description here (such as "the 8-cell stage").

-Could the authors please clarify what is shown in Figure 1C? Percent of Nr5a2 peaks that are within a certain class of repeat? If so, shouldn't a column in that table add up to 100%? Moreover, could it be that there are the same number of peaks at B1/Alu elements, for example, across different stages and that the percent of peaks only shows a decrease because of an increase in total peak number?

-Line 140: instead of "by promoting the Hippo signaling pathway" consider "by promoting the Hippo signaling pathway and polarization"

- The C1-4 clusters in Figure 2E are not sufficiently explained.
- Line 204: "We next examined genome-wide transcriptional changes regulated by NR5A2 and KLF5 (Figure S4A and Table S3)." What embryonic stage was this analysis performed at?
- Line 212: "We cannot exclude that the minor effect of Klf5 KD alone is due to inefficient knockdown (Figure 2B)." It is not clear how this is referring to Figure 2B.

Reviewer 2

Kobayashi and colleagues set out to investigate how the pioneer transcription factor NR5A2 influences chromatin reprogramming during the transition from totipotency to pluripotency in mouse embryos, which occurs between the 2C and morula stages of development. They initially mapped NR5A2 binding using CUT&Tag methods across this developmental window using in vitro cultured embryos. They found that NR5A2 binds the genome most extensively at the 8-cell stage, and binding sites decrease at the morula stage. They also found that NR5A2 binds to LINE elements at early stages, and then binding shifts to promoters in morula. They also used ATAC-seq and C&T to measure accessibility and H3K27ac. They found that NR5A2 binds preferentially at promoters and putative enhancers from 2C to morula stages, which implies that NR5A2 is involved in transcriptional activation. KLF and GATA motifs were abundant in NR5A2 peaks at the 8-cell and morula stages, leading them to investigate Klf5 and Gata6. Through combined loss-of-function, genomics, and immunofluorescence studies, they found that NR5A2 activates Klf5 and Gata6, and that NR5A2 stimulates chromatin accessibility, while KLF5 enhances H3K27ac levels when co-bound. Finally, through in vitro biochemical studies on recombinant nucleosomes, they found that NR5A2 binds nucleosomes together with KLF5 and GATA6, supporting simultaneous chromatin engagement. They propose a model in which a feed-forward loop driven by NR5A2 activates lineage-specifying factors, and co-binding of these factors reinforces gene activation during early development. I found the rationale for this study to be reasonable, the results to be solid, and their conclusions to be mostly supported. I have only a few major concerns, along with some suggestions.

Major concerns:

1. The authors make the claim that NR5A2 regulates KLF5 and GATA6, and through this regulation, combined with cooperative function, these TFs regulate early development. Their in vitro embryo survival results following KD support these conclusions, but their genomic studies indicate that NR5A2 is much more important than KLF5. Why is this? Does KLF5 do something else to embryos that they didn't measure? Is there some degree of inefficiency in their assays. An alternative explanation is that KLF5 is only functionally important in the absence of NR5A2.
2. The authors claim, based on in vitro assays and correlative genomics studies, that NR5A2 binds chromatin at the same time as KLF5/GATA. This claim would be better supported with the addition of ChIP-ChIP-Westerns, ChIP-ChIP-seq assays, or ChIP-MassSpec assays.
3. There is an overall lack of statistical analysis throughout the manuscript when comparing changes in chromatin features, as measured by genomics methods, such as the H3K27ac or accessibility measures. In order to conclude that these chromatin features do change (such as under KD conditions), the results need to be supported by a statistical test. Performing DiffBind analysis would be best in this case.
4. The authors' focus on Xist regulation is confusing, distracting, tangential, and somewhat weakly supported. The study would be made stronger if this portion of the manuscript were removed.

Minor concerns and suggestions:

5. Further CUT&Tag experiments on KLF5 under NR5A2 KD conditions, and vice versa, can help strengthen their conclusion that NR5A2 directly regulates KLF5 binding/activity.
6. The authors observed that KLF and GATA motifs are enriched in NR5A2 peaks at the 8-cell and morula stages, but they did not compare the motif enrichment at earlier stages. Do the motifs identified differ?
7. The authors identified that many NR5A2 peaks overlap with SINE B1 at the 2-cell stage and this overlap gradually decrease during development. They analyzed gene expression after single knockdown of NR5A2 and double knockdown of NR5A2 and KLF5, but they did not investigate LINE B1 elements. Further analysis of repetitive loci would be very informative.

8. I am left wondering how NR5A2 becomes relocalized from LINE B1 elements to promoters during the transition from 2C to morula. Do B1 elements become less accessible or gain H3K9me3 when NR5A2 leaves these loci?

9. How were "repressive" regions defined in Fig 1B? This is especially relevant because in Fig 1C they find that 70% of peaks are at repetitive sites.

10. In Fig 5, it seems that KLF5 KD causes an increase in Gata6. Is this the case? If so, this would be an interesting outcome worth following up on.

Signed - Patrick Murphy

Reviewer 3

SUMMARY OF THE ADVANCE MADE IN THIS PAPER AND ITS POTENTIAL SIGNIFICANCE TO THE FIELD

The manuscript by Kabayashi et al. builds on prior data identifying essential functions for the nuclear hormone receptor NR5A2 in early mouse development by demonstrating the connection between this pioneer factor and additional transcription factors (KLF5 and GATA6) that regulate development. The authors outline a mechanism by which regulatory networks can feed forward to promote differentiation, which is likely to be of broad interest. In general, the experiments are rigorously performed. The role of NR5A2 in promoting expression of KLF5 and GATA6 is convincing, but the data supporting the co-regulation of NR5A2 by these factors is much less robust. This results in many instances throughout the manuscript where the interpretation is overstated. In addition, some contradictions with prior literature are not clearly discussed.

SUGGESTIONS TO AUTHORS

Major critiques:

1. In many of the genomic comparisons, the authors do not use statistical analyses to robustly call differences between experiments. For example, with the ATAC-seq and H3K27ac CUT&Tag the authors should use DESeq (or equivalent) to determine the statistically significant changes in accessibility and acetylation. This is important for the data in Figures 4 and 5 where many of the changes highlighted in the Genome Viewer snapshot do not appear particularly robust. Are these called as significant using rigorous comparisons that consider variability between replicates? (In fact, the authors comment that the tracks shown are averages of the replicates since there was variability.) This need for statistical measurements of differences is notable in many instances but is exemplified in Figure 4D. The most robust changes in ATAC-seq and H3K27ac signal upon depletion of NR5A2 are in peaks that are lowly occupied by the factor. This is counterintuitive as one might predict that regions that depend most strongly on the factor would be those with the robust CUT&Tag signal. It is possible that these lower peaks, which also show the lowest signals for ATAC-seq and H3K27ac, are the most variable between replicates and are therefore the most subject to minor differences. Using more statistically robust means of calling differences and experiments to confirm that the differences observed are biological and not technical (such as spike in normalization, etc) is important.

2. The explanation for the focus on the GATA and KLF families is not clear from the manuscript as written. Indeed, it would be important to note prior work in this context. For example, in Festuccia et al. Science 2024 it is suggested that KLF transcription factors might function with NR5A2. Similarly, the authors need to address other data from this manuscript that appears to contradict their study. In Figure 3 of Festuccia et al., GATA6 levels in E2.75 embryos that are mutant for maternal and zygotic NR5A2 appear to have normal levels of GATA6. This seems to contradict the data shown in Figure 5B where GATA6 levels are decreased upon RNAi targeting Nr5a2. The authors must address this discrepancy.

3. The data presented in Figure 6 G-I do not obviously support the claimed co-binding of NR5A2, KLF5 and GATA6 on the nucleosomes. The migration of the complexes on the gel is not clearly different when incubated with additional factors, and the decrease in free nucleosome is possibly explained by binding individual proteins. Are there conditions in which the resolution of the multiply bound nucleosome could be better resolved? This would be important for supporting the claim.

4. Some of the data are challenging to interpret given the complex interactions between these factors. The single and double knockdown experiments are nice but hard to interpret given the effect of Nr5a2 knockdown on KLF5 levels. The evidence that KLF5 and GATA6 regulate NR5A2 (as claimed on line 351) is not particularly convincing and would be strengthened by testing the binding of NR5A2 in the knockdown of KLF5 or GATA6. Similarly, the redundancy between KLF and GATA family members makes simple conclusions difficult. We encourage the authors to tone down some of their conclusions. As an example, the heading on line 250 should be modified. The data show that KLF5 binds to the promoters of other family members and that their expression changes in the single knockdown (but notably not the double knockdown). As such, the data do not convincingly show that "KLF5 directly regulates other KLF family members." Similar arguments about direct regulation are made for NR5A2 on line 275 that should be modified to take into accounts that some of these effects may be indirect despite binding of the given factor. Statements in the abstract are also over interpreted. For example, the data supporting the cooperation of KLF5 with NR5A2 in H3K27ac is not very strong and therefore the statement on line 26 that "KLF5 cooperates with NR5A2 to enhance H3K27ac deposition" should be modified.

Other critiques:

1. Clarity would be improved if the authors were more explicit about how hours post fertilization correspond to the various stages (2C, 4C, 8C, morula). Similarly, in the introduction it would be useful to connect the morula stage to the blastocyst and the differentiation of ICM and TE.
2. Figure 1D is surprising. It appears that most of the NR5A2 binding at SINE B1 elements changes between stages with ~70% gained and lost between the 2C-4C and over 50% between the 4C-8C. The authors focus on the gain (line 113), but this seems misplaced as there are nearly equally dramatic changes in gains and losses between stages.
3. The authors make a strong claim concerning GO term analysis and the function of cluster 3 in stem cell maintenance in the results. This text would be better suited for the discussion.
4. Can the authors define hatching as discussed on line 193?
5. On line 303, the authors might substitute "in principle" for "principally". The first suggests the possibility. The second makes it sound like this is the major form of binding.
6. In the legend for Figure 2A, please specify to what the comparison was made to identify enrichment.
7. In the Figure 4 legend - the second 'D' (in line 490) should be 'E'.
8. The model in Figure 7B would be strengthened if the authors related the molecular events to developmental timepoints.

First revision

Author response to reviewers' comments

We would like to sincerely thank all reviewers for their positive comments and constructive suggestions, which we have addressed in the revised manuscript. The new parts are highlighted in yellow. The revision experiments and further analyses suggested by reviewers have greatly strengthened the manuscript. In brief,

- 1) As suggested the Reviewer 1, we determined the GATA1 chromatin binding profile using CUT&RUN to further investigate how NR5A2 might regulate *Xist* expression. GATA1 occupancy was also detected on regulatory elements RE79 and RE97 that control *Xist* expression. This result provides further support for a model in which NR5A2 regulates *Xist* expression, either directly or indirectly, through its role in up-regulating GATA factor expression (**new Figures**

5D and S8).

- 2) As suggested by Reviewer 1, we examined the abundance of the first lineage cell marker NANOG (for inner cell mass) and CDX2 (for trophectoderm) using immunofluorescent staining in 16-cell stage embryos. We found that the protein abundance of both first lineage markers, NANOG and CDX2, were markedly decreased upon *Nr5a2 Klf5* double KD (new Figure S3B,C). This analysis may explain the severe phenotype of *Nr5a2 Klf5* double KD embryos and supports a role of NR5A2 and KLF5 prior to the first cell lineage segregation.
- 3) As suggested by Reviewers 2 and 3, we performed differential binding analysis for ATAC-seq and H3K27ac CUT&Tag. Differential binding analysis of H3K27ac CUT&Tag detected 158 and 1,348 regions with decreased H3K27ac signals upon *Nr5a2* KD and *Nr5a2 Klf5* KD, respectively. In contrast, *Klf5* KD had negligible effects on H3K27ac, consistent with the mild transcriptional changes detected by RNA-seq data. This result suggests a synergistic role of NR5A2 and KLF5 in H3K27ac deposition (new Figure 4C,D) and could explain the stronger developmental phenotype of the double knockdown embryos.
- 4) Our *in vitro* data suggest that NR5A2, KLF5 and GATA6 can co-occupy nucleosomes, but whether this is the case *in vivo* is not known. To address this, we performed a proximity ligation assay (PLA, Soderberg et al., 2006) by immunofluorescence. In brief, the assay detects the close physical proximity of two target proteins, where oligonucleotide-conjugated antibodies in proximity generate a quantifiable rolling-circle amplification signal. Using the PLA assay, we obtained evidence that NR5A2 and KLF5 are in close proximity in 8-cell stage embryos (new Figure 4F,G), consistent with the *in vitro* data.
- 5) Following the comment by Reviewer 1, we addressed the correlation between transcriptional activation and NR5A2 binding. We found that down-regulated genes upon *Nr5a2* KD and *Nr5a2 Klf5* dKD were more enriched for NR5A2 and KLF5 peaks in their neighboring distal regions compared with up-regulated genes (new Figure S4B), supporting direct activation role in NR5A2 and KLF5. In addition, both *SINE B1/Alu*- and non-*SINE B1/Alu*-bound NR5A2 peaks also preferentially located near down-regulated genes upon *Nr5a2 Klf5* dKD (new Figure S4C). These results suggest that NR5A2 regulate gene expression through binding to both *SINE B1/Alu* and non-*SINE B1/Alu* regions.
- 6) To streamline the message of our manuscript, we removed the section "KLF5 directly regulates other KLF family members" from the revised manuscript since this was not investigated further.

Below, we provide detailed responses to each comment.

Reviewer 1: SUMMARY OF THE ADVANCE MADE IN THIS PAPER AND ITS POTENTIAL SIGNIFICANCE TO THE FIELD

In this manuscript Kobayashi et al investigate how the pioneer TF Nr5a2 functions during early mouse development, an interesting question for the field of developmental biology. They characterize genome-wide binding profiles for Nr5a2 at different stages of preimplantation development and identify KLF and GATA factor motifs enriched withing Nr5a2 peaks, suggesting that these factors co-regulate active enhancers with Nr5a2. They follow up with functional experiments on the roles of Nr5a2, Klf5 and their combined effect on preimplantation development, revealing that the two factors function synergistically in blastocyst formation. They also use an *in vitro* assay to show that Nr5a2, Klf5 and Gata6 can bind to the same nucleosome, supporting the model that these factors can act together at the same site. There is a wealth of experimental data in this manuscript, but the story is somewhat incoherent in places.

We are very grateful to the reviewer for highlighting our findings and would like to thank the reviewer for providing constructive suggestions.

SUGGESTIONS TO AUTHORS

Major:

-The claim that Nr5a2 regulates Klf5 and Gata6 is not supported. Klf5 transcript and protein are indeed down upon Nr5a2 KD, but whether Nr5a2 binds in the vicinity of Klf5 and whether this binding site changes (accessibility or H3K27ac) upon Nr5a2 KD is not examined.

We thank the reviewer for pointing this out. We focused on the *Klf5*-adjacent distal regulatory element that controls *Klf5* expression in mouse embryonic stem cells, recently annotated as K5aSE (Su et al., 2025). K5aSE deletion led to impaired proliferation, differentiation, and *Klf5* expression (Su et al., 2025). We detected NR5A2 binding at this regulatory element and observed a reduction of H3K27ac levels upon *Nr5a2* KD at the 8-cell stage (new Figure S6A). These findings support a role for NR5A2 in regulating *Klf5* expression during murine pre-implantation development.

The direct regulation of *Gata6* by Nr5a2 is not evident. *Gata6* transcripts are negligibly downregulated in Nr5a2 KD and it is difficult to comprehend how the mild down-regulation of *Gata6* protein is due to Nr5a2 KD. Would Nr5a2 regulate *Gata6* post-transcriptionally? Also, Figure 5D shows Nr5a2 binding at a site near *Gata6*, but in the KD chromatin accessibility and H3K27ac do not change at this site. In sum, regulation of *Klf5* needs more evidence and regulation of *Gata6* is not supported by the data.

The down-regulation of *Gata6* expression upon *Nr5a2* KD is consistent with a previous study (Lai et al., 2023, *Cell Research*). We used the same siRNA sequence and concentration to knockdown *Nr5a2*; however, the KD efficiency of *Gata6* RNA and protein levels was milder than previously reported. The difference could be technical or biological, e.g. the studies used different mouse strains. To obtain zygotes, Lai et al. used female C57BL/6J and male PWK/PhJ strains, whereas we used F1 hybrid mice, female B6129D1 and male B6CBAF1.

We apologize for the confusion. The previous Figure 5D showed chromatin accessibility and H3K27ac near the *Xist* site, but not the *Gata6* gene. We showed the reduction of H3K27ac upon *Nr5a2* KD on the distal region of *Gata6* gene (now Figure S7A). Hence, these data support a role for NR5A2 in regulating GATA6 expression.

- Along the same lines, this raises the concern that the proposed Nr5a2-Gata6 link for X inactivation are not supported by the data. The down-regulation of *Gata1* transcript and the potentially direct role of Nr5a2 in this better supported. However, the authors do not examine the binding of *Gata1* in their study.

We thank the reviewer for suggesting a valuable experiment. Following the reviewer's suggestion, we examined GATA1 chromatin binding profile at the 8-cell stage initially using CUT&Tag. However, the GATA1 CUT&Tag profile is sparse compared with the GATA6 CUT&Tag (Figure R1), which is potentially due to the low protein expression level of GATA1 or a less stable chromatin association. Our CUT&Tag protocol includes a fixation step to stabilize transcription factor binding, but it did not help in this case. Since CUT&Tag includes a washing step with relatively high salt (300 mM NaCl), we reasoned that stringent washing might reduce GATA1 chromatin binding detection. We therefore performed CUT&RUN as an alternative approach and successfully improved the resolution of GATA1 chromatin binding profile at the 8-cell stage (Figure R1 and new Figure S8). GATA1 occupancy was also detected on regulatory elements RE79 and RE97 that control *Xist* expression (new Figure 6D). We also observed a reduction of H3K27ac at the transcription start site of the *Gata1* gene upon *Nr5a2* KD (Figure S7A). This additional data further supports a model in which NR5A2 regulates *Xist* expression, either directly or indirectly, through its role in up-regulating GATA factor expression. As ATAC-seq and H3K27ac CUT&Tag data in *Nr5a2* KD do not add much to the interpretation of *Xist* regulation, we removed these from the previous Figure 6D.

Figure R1: IGV snapshot showing GATA1 CUT&RUN and CUT&Tag on the 8-cell stage

-The section on "KLF5 directly regulates other KLF family members" seems like an afterthought and does not add much to the main message of the paper. It is also unclear how this function of Klf5 relates to Nr5a2.

Thank you for the comment. We agree that this paragraph does not add much to our message. We had a similar comment from Reviewer 3. To streamline the message of our manuscript, we removed this paragraph.

-The analysis of phenotypes in KDs (Figure 3) is quite rudimentary. It is sufficient to make the claim of cooperative action between Nr5a2 and Klf5, however, it would be a lot more informative if the authors could perform lineage-specific analysis, including immunofluorescent stainings for cell fate markers and cell counts for each cell type. This could be particularly interesting as the authors later show that genes involved in the first cell fate decision are affected in KDs.

We thank the reviewer for suggesting this important additional analysis. Because *Nr5a2 Klf5* dKD embryos cause a severe developmental failure and show fragmentation and dying before reaching to the blastocyst stage (Figures 3E and 3F), we fixed embryos at the 16-cell stage (early morula, 66-68 hpf) and examined the level of the first lineage cell marker NANOG (for inner cell mass) and CDX2 (for trophectoderm) by immunofluorescent staining. We found that the protein abundance of both first lineage markers, NANOG and CDX2, was markedly decreased upon *Nr5a2 Klf5* double KD (new Figure S3B), indicating impaired first lineage segregation. This result may explain the severe phenotype of *Nr5a2 Klf5* double KD embryos and supports a role of NR5A2 and KLF5 in regulating ICM and TE genes.

Minor:

-It is somewhat unclear to me if Nr5a2 is only acting at repetitive elements or also at other genomic sites?

We thank the reviewer for the comment. To determine whether transposable element (TE)-bound and non TE-bound NR5A2 contribute to gene regulatory function, we asked if the binding sites in each group located within the proximity of genes that are affected by *Nr5a2 Klf5* dKD. We first considered comparing peaks that are binding at *SINE B1/Alu* loci, as these are the main TE family that bound by NR5A2 in early development (Figure 1 D). Around 30% of genes down-regulated upon *Nr5a2 Klf5* dKD have NR5A2 binding in their upstream regions (new Figure S4C). In contrast, the a much smaller proportion of up-regulated and randomly selected genes contains NR5A2 binding in their upstream regions, suggested that *SINE B1/Alu*-bound NR5A2 contributes to transcriptional regulation of these genes (new Figure S4C). While only approximately 10% of down-regulated genes have non-*SINE B1/Alu* bound NR5A2 in their upstream regions, the occurrence of binding is still higher than those from up-regulated and random genes (new Figure S4C). The general ratio of down-regulated genes with non-*SINE B1/Alu* NR5A2 binding could partially be explained by lower number of peaks in this group (Figure 1D). Taken together, these data suggest that both *SINE B1/Alu* bound- and non-*SINE B1/Alu*- NR5A2 peaks regulate gene expression in 8-cell embryos.

In addition, we performed the analysis to NR5A2 binding sites that overlap and not overlap with any TE regions (Figure R2). As expected, we observed similar trend for TE-bound NR5A2. The results from non-TE bound NR5A2 also showed similar trend, however, the proportion of differentially expressed genes containing peaks from this group is much smaller as only 6.6% of all NR5A2 binding sites in 8-cells are non-TE binding.

Figure R2: Cumulative percentages of differentially expressed genes upon *Nr5a2* and *Klf5* dKD

-Please specify what morula is. Is it referring to compacted 8 and 16-cell stage embryos? Whereas "8-cell stage" is referring to uncompact 8-cell embryos?

We refer to uncompact 8-cell and a ~32-cell stage embryos without fluid-filled cavity as "8-cell" and "morula", respectively. We have provided the specifications of each cell stage in the material and methods section, "collection of mouse embryos" (lines 589-590).

-Consider moving Figure S1C to main figure, as this is an important overview of *Nr5a2* binding dynamics across embryo stages.

We moved the old Figure S1C to the **main Figure 1B**, according to your suggestion.

-Line 116 "until cleavage-stage blastomeres" - as far as we know cleavage divisions occur until the blastocyst stage, so it would be preferred to use a more concrete description here (such as "the 8-cell stage").

We amended the text accordingly (lines 116-117).

-Could the authors please clarify what is shown in Figure 1C? Percent of *Nr5a2* peaks that are within a certain class of repeat? If so, shouldn't a column in that table add up to 100%? Moreover, could it be that there are the same number of peaks at *B1/Alu* elements, for example, across different stages and that the percent of peaks only shows a decrease because of an increase in total peak number?

We apologize for the confusion. As you mentioned, the percentage indicates the proportion of *NR5A2* peaks that overlap with repetitive elements. The sum of each column would add up to 100% only when each *NR5A2* peak overlaps with at most one family of repeat elements. However, as shown in **Figure R3A**, it is common for one peak to overlap with multiple elements from different families. For this reason, each column does not add up to 100%. We added this detail in the figure legend (lines 464-466).

The exact number of *NR5A2* peaks at *B1/Alu* in each stage is not the same at each cell stage. (**Figure R3B**). However, the 4-cell and morula stages contain similar number of *B1*-bound *NR5A2* peaks, but the percentage of *B1*-bound loci is smaller in morula as it contains more total number of peaks.

Figure R3: (A) IGV snapshots of multiple loci where NR5A2 peaks overlap with more than one repeat elements from different families. (B) Total number of NR5A2 peaks in each stage of development. Proportion of B1-bound peaks is indicated in black.

-Line 140: instead of "by promoting the Hippo signaling pathway" consider " by promoting the Hippo signaling pathway and polarization"

We amended the text accordingly (line 139).

-The C1-4 clusters in Figure 2E are not sufficiently explained.

We thank the reviewer for pointing this out. We have revised this section to better explain the characteristic of each cluster of NR5A2 peaks at morula stage (lines 169-178). In brief, we added the general description that C1 and 2 are loci that are already bound by NR5A2 since the 2-cell stages, while the other are newly bound loci. We also mentioned that constant NR5A2 bound loci, especially C2, tends to overlap with *SINE B1/Alu*. In contrast, the new peaks, especially C3 that co-occupied by the three TFs, tends to be non-*SINE B1* loci. While the *SINE B1/Alu* density can be inferred from the heatmap (Figure 2E), the exact percentage of NR5A2 peaks in each cluster that overlap with *SINE B1/Alu* is provided in the new Figure S2D to better support the results.

-Line 204: "We next examined genome-wide transcriptional changes regulated by NR5A2 and KLF5 (Figure S4A and Table S3)." What embryonic stage was this analysis performed at?

We added "the 8-cell stage" in the text (line 213).

-Line 212: "We cannot exclude that the minor effect of Klf5 KD alone is due to inefficient knockdown (Figure 2B)." It is not clear how this is referring to Figure 2B.

We are sorry about for the confusion caused by a typo. We amended the text to "Figure 3B-D" (line 218).

Reviewer 2: Kobayashi and colleagues set out to investigate how the pioneer transcription factor NR5A2 influences chromatin reprogramming during the transition from totipotency to pluripotency in mouse embryos, which occurs between the 2C and morula stages of development. They initially mapped NR5A2 binding using CUT&Tag methods across this developmental window using in vitro cultured embryos. They found that NR5A2 binds the genome most extensively at the 8-cell stage, and binding sites decrease at the morula stage. They also found that NR5A2 binds to LINE elements at early stages, and then binding shifts to promoters in morula. They also used ATAC-seq and C&T to measure accessibility and H3K27ac. They found that NR5A2 binds preferentially at promoters and putative enhancers from 2C to morula stages, which implies that NR5A2 is involved in transcriptional activation. KLF and GATA motifs were abundant in NR5A2 peaks at the 8-cell and morula stages, leading them to investigate Klf5 and Gata6. Through combined loss-of-function, genomics, and immunofluorescence studies, they found that NR5A2 activates Klf5 and Gata6, and that NR5A2 stimulates chromatin accessibility, while KLF5 enhances H3K27ac levels when co-bound. Finally, through in vitro biochemical studies on recombinant nucleosomes, they found that NR5A2 binds nucleosomes together with KLF5 and GATA6, supporting simultaneous chromatin engagement.

They propose a model in which a feed- forward loop driven by NR5A2 activates lineage-specifying factors, and co-binding of these factors reinforces gene activation during early development. I found the rationale for this study to be reasonable, the results to be solid, and their conclusions to be mostly supported. I have only a few major concerns, along with some suggestions.

We sincerely thank the reviewer for the positive evaluation of our work and the insightful comments.

Major concerns:

1. The authors make the claim that NR5A2 regulates KLF5 and GATA6, and through this regulation, combined with cooperative function, these TFs regulate early development. Their *in vitro* embryo survival results following KD support these conclusions, but their genomic studies indicate that NR5A2 is much more important than KLF5. Why is this? Does KLF5 do something else to embryos that they didn't measure? Is there some degree of inefficiency in their assays. An alternative explanation is that KLF5 is only functionally important in the absence of NR5A2.

We thank the reviewer for this important comment. One possible explanation is that KLF5 requires NR5A2 chromatin binding on regions targeted by both, thereby recruiting histone acetyltransferase to promote active chromatin. Since KLF5 expression is regulated by NR5A2, we could not directly test whether NR5A2 facilitates KLF5 chromatin binding by the siRNA-mediated knockdown approach. An acute protein depletion approach for short-term perturbation is required to dissect the interplay of these TFs in the future.

There is also the possibility that there is some degree of inefficiency in the KD assay. KLF5 gross abundance is decreased upon KD. However, there are precedents for chromatin-bound proteins that the actual chromatin-bound fraction can be difficult to deplete, e.g. auxin-mediated degradation of CTCF results in a gross loss of CTCF but not at CTCF-bound sites (Luan et al., Cell Reports, 2021). In this scenario, the double KD would have a stronger effect because the synthesis of KLF5 is already strongly reduced.

Based on the new data, we have extensively revised the manuscript and discussed these considerations in lines 380-386 and "Study limitations".

2. The authors claim, based on *in vitro* assays and correlative genomics studies, that NR5A2 binds chromatin at the same time as KLF5/GATA. This claim would be better supported with the addition of ChIP-ChIP-Westerns, ChIP-ChIP-seq assays, or ChIP-MassSpec assays.

We appreciate the reviewer's comment, and we fully agree that experiments are needed to investigate whether NR5A2 binds chromatin at the same time as KLF5/GATA6 *in vivo*. The approaches suggested by the reviewer would indeed be the gold standard to address this question. Unfortunately, due to the extremely limited material (10-20 embryos per mouse) and the low abundance nature of transcription factors, these ChIP-based assays are currently technically not feasible in this system. To nevertheless address this open point, we turned to a proximity ligation assay in which the close physical proximity of target proteins via antibodies conjugated to oligonucleotides facilitates a rolling circle amplification signal that can be quantified (Soderberg et al., 2006). Using this assay, we tested pair-wise proximity of NR5A2 with KLF5 and NR5A2 with GATA6 using 8-cell embryos. We found that the number of PLA signals detected with NR5A2 and GATA6 antibodies was lower than that observed with NR5A2 and KLF5 antibodies (**new Figure 4F,G**). We are thus less confident about the NR5A2-GATA6 data and include it as **Figure R4**. The signals for NR5A2 and KLF5 are more robust and suggest that these transcription factors are in close proximity, consistent with the result of co-occupancy of TFs determined by CUT&Tag (Figure 2D). These results provide evidence that NR5A2 is in close proximity with KLF5 and potentially GATA6 in the nucleus, supporting the results of *in vitro* assays.

Figure R4: Proximity ligation assay with NR5A2 and GATA6 antibodies. (A) A representative nucleus with z-stack is shown in each condition. Scale bars, 10 μm . (B) Quantification of PLA signal density in each condition from two independent experiments. P values (Mann-Whitney test) are shown. Sample sizes (embryos) in each replicate are as follows: no primary, n = 6, 6; α -NR5A2, n = 6, 6; α -GATA6, n = 6, 8; α -NR5A2 + α -GATA6, n = 15, 15.

3. There is an overall lack of statistical analysis throughout the manuscript when comparing changes in chromatin features, as measured by genomics methods, such as the H3K27ac or accessibility measures. In order to conclude that these chromatin features do change (such as under KD conditions), the results need to be supported by a statistical test. Performing DiffBind analysis would be best in this case.

We thank the reviewer for this critical suggestion. We have performed differential binding analysis by DiffBind as suggested. Our analysis identified 158 and 1,348 regions that showed decrease in H3K27ac upon *Nr5a2* KD and *Nr5a2 Klf5* KD conditions, respectively (new Figure 4C). Following this, we added a heatmap showing enrichment of H3K27ac on gain and loss of regions identified by differential binding analysis (new Figure 4D). We observed a more reduction of H3K27ac in at *Nr5a2 Klf5* KD at the NR5A2 and KLF5 co-occupied regions (new Figure 4D), which supports our claim.

The statistical power of differential binding analysis was not as powerful in our ATAC-seq datasets as there were only 2 replicates per conditions. We observed inconsistency of signals between the two replicates of ATAC-seq data in the non-targeting siRNA control conditions, which could partially explain the small number of affected regions detected in differential binding analysis of ATAC-seq dataset (Figure R5). Nevertheless, chromatin accessibility showed a similar trend in regions where H3K27ac is affected (new Fig. S5D), and a reproducible reduction of chromatin accessibility near down-regulated genes by *Nr5a2* KD is observed (new Figure S5F). The observations of decreased accessibility are consistent with previous studies (Lai et al., 2023; Festuccia et al., 2024). To ensure accuracy in data representation, we removed merged replicate ATAC-seq data from the analysis.

Figure R5: Differential binding analysis of ATAC-seq

4. The authors' focus on *Xist* regulation is confusing, distracting, tangential, and somewhat weakly supported. The study would be made stronger if this portion of the manuscript were removed.

Thank you very much for your comment. We agree that the regulation of *Xist* deserves further study; however, we believe that our current findings provide meaningful insights into this mechanism. Following Reviewer 1's suggestions, we have provided the chromatin binding profile of GATA1, in which the transcript is regulated by NR5A2 (new Figure S8). We found that GATA1 occupancy is also detected on regulatory elements RE79 with NR5A2 occupancy and RE97 (new Figure 6D). This result provides evidence showing that GATA factor (GATA1 and GATA6) targets *Xist* regulatory elements during murine pre-implantation development, supporting previous work (Ravid Lustig et al., 2023). Together, our model extends the transcriptional regulatory network regulated by NR5A2 linked to *Xist* expression by regulating GATA factors. Therefore, we believe that the section would be of interest to the fields of X chromosome inactivation and early developmental gene regulation.

Minor concerns and suggestions:

5. Further CUT&Tag experiments on KLF5 under NR5A2 KD conditions, and vice versa, can help strengthen their conclusion that NR5A2 directly regulates KLF5 binding/activity.

We thank the reviewers for the suggestion. As NR5A2 regulates KLF5 expression at both RNA and protein levels, it would be difficult to disentangle the effect of *Nr5a2* KD on KLF5 binding through reduced protein abundance versus a reduced interaction with NR5A2. We were thus primarily able to examine the NR5A2 binding profile upon *Klf5* KD. To increase the sensitivity of chromatin binding profiles with a limited number of input cells, NR5A2 Targeted insertion of promoters sequencing (TIP-seq) (Bartlett et al., 2021) was performed (control 8-cell embryos, rep1: 25, rep2: 26. *Klf5* KD 8-cell embryos, rep1: 30, rep2: 30). *Klf5* KD efficiency was confirmed using the residual embryos (Figure R6). This experiment showed that NR5A2 occupancy is not significantly altered upon *Klf5* KD, suggesting that NR5A2 binding is largely independent of KLF5 binding (new Figure S6).

Figure R6: Klf5 knockdown efficiency for TIP-seq experiment

6. The authors observed that KLF and GATA motifs are enriched in NR5A2 peaks at the 8-cell and morula stages, but they did not compare the motif enrichment at earlier stages. Do the motifs identified differ?

We apologize for the lack of clarity. The motif enrichment analysis in Figure 2A used the same set of reference transcription motifs from the database for each set of NR5A2 peaks across developmental stages. The Figure shows that KLF and GATA motifs become more co-enriched with NR5A2 binding sites in the later stages. However, it does not rule out the co-enrichment of these motifs in earlier stages. For example, KLF motifs are also enriched in NR5A2 peaks at 2-cell and 4-cell stages. The level of enrichment, indicated by the odds ratio and p-value, becomes higher at NR5A2 binding sites at later stages. We have added the details in the figure legends (lines 474-476).

7. The authors identified that many NR5A2 peaks overlap with SINE B1 at the 2-cell stage and this overlap gradually decrease during development. They analyzed gene expression after single knockdown of NR5A2 and double knockdown of NR5A2 and KLF5, but they did not investigate SINE B1 elements. Further analysis of repetitive loci would be very informative.

We thank the reviewer for this suggestion. We performed differential expression analysis of transposable elements from our datasets (**Figure R7**). We observed a small but statistically not significant decrease in expression of *B1* families that are normally expressed in mouse embryos (*B1_Mus1*, *B1_Mus2*, and *B1_Mm*). Knockdown of *Nr5a2* and/or *Klf5* also minimally affected expression of other repeat elements.

Figure R7: Differential expression analysis of repeat element families upon *Nr5a2* and/or *Klf5* knockdown conditions. Repeat element families from *SINE B1/Alu* class are highlighted by purple circle.

8. I am left wondering how NR5A2 becomes relocalized from SINE B1 elements to promoters during the transition from 2C to morula. Do B1 elements become less accessible or gain H3K9me3 when NR5A2 leaves these loci?

We appreciate the review for suggesting insightful comments. We examined the change of chromatin modifications and accessibility at NR5A2 binding sites that overlapped with *SINE B1/Alu* (**Figure R8**). Interestingly, we found that NR5A2 binding pattern at each developmental stages correlated with chromatin accessibility and active histone modification marks, H3K27ac (**Figure R8**). Genomic sites that are bound early on by NR5A2 and are lost during development also show a decrease in chromatin accessibility during development. Whether NR5A2 directly regulates chromatin accessibility at these loci remains to be determined. In contrast to active chromatin marks, repressive chromatin features show similar patterns regardless of the stage that NR5A2 binds to these loci. For example, *SINE B1/Alu* loci binds by NR5A2 in both early (cluster 1, 2, and 4) or late (cluster 3, and 5) gain similar levels of H3K9me3 and H3K27me3 from the 8-cell stage onward.

We used the following public data for this analysis: GSE153496 (Mei et al., Nat. Genet., 2021), GSE158360 (Chen et al. Nat. Genet., 2021), GSE98149 (Wang et al., Nat. Cell Bio., 2018), and GSE73952 (Liu et al., Nature, 2016).

Figure R8: Chromatin environment at NR5A2 peaks with *SINE B1/Alu* at different stages of preimplantation development.

9. How were "repressive" regions defined in Fig 1B? This is especially relevant because in Fig 1C they find that 70% of peaks are at repetitive sites.

We applied ChromHMM (Ernst and Kellis, 2017) to annotate chromatin states from chromatin accessibility and other various histone modification data in each cell stage. A repressive chromatin state is defined as regions enriched with H3K9me3 and H3K27me3. Please see Figure S1D and material methods (lines 892-901) for more details.

10. In Fig 5, it seems that *KLF5* KD causes an increase in *Gata6*. Is this the case? If so, this would be an interesting outcome worth following up on.

Thank you for the valuable comment. We examined the GATA6 protein expression by immunofluorescent staining in *Klf5* KD embryos. Consistent with the RNA-seq result, GATA6 abundance is slightly increased upon *Klf5* KD (Figure R9A). However, H3K27ac levels near the *Gata6* gene were not altered (Figure R9B). Therefore, it is difficult to infer a direct link between GATA6 and *KLF5* from the current data.

Figure R9: (A) Representative images (left) and quantifications (right) of immunostaining analysis showing KLF5 (green), GATA6 (grey), and DAPI (cyan) of 8-cell embryos. NR5A2 and KLF5 signals are shown in mid- section images. DAPI signals are presented as full z-stack images to visualize all nuclei. The number of embryos examined (n) from two independent experiments is indicated. Scale bars, 20 μ m. Bars overlaid on the plots indicate means. P values (t-test, two-sided) are shown. (B) IGV snapshot showing merged replicate H3K27ac CUT&Tag profile near *Gata6* gene.

Signed - Patrick Murphy

Reviewer 3: SUMMARY OF THE ADVANCE MADE IN THIS PAPER AND ITS POTENTIAL SIGNIFICANCE TO THE FIELD

The manuscript by Kabayashi et al. builds on prior data identifying essential functions for the nuclear hormone receptor NR5A2 in early mouse development by demonstrating the connection between this pioneer factor and additional transcription factors (KLF5 and GATA6) that regulate development. The authors outline a mechanism by which regulatory networks can feed forward to promote differentiation, which is likely to be of broad interest. In general, the experiments are rigorously performed. The role of NR5A2 in promoting expression of KLF5 and GATA6 is convincing, but the data supporting the co-regulation of NR5A2 by these factors is much less robust. This results in many instances throughout the manuscript where the interpretation is overstated. In addition, some contradictions with prior literature are not clearly discussed.

We sincerely thank this reviewer for their supportive assessment of our study. We greatly appreciate the deep reading of the manuscript and the numerous constructive comments that have helped to improve the quality and rigour of the manuscript.

SUGGESTIONS TO AUTHORS

Major critiques:

1. In many of the genomic comparisons, the authors do not use statistical analyses to robustly call differences between experiments. For example, with the ATAC-seq and H3K27ac CUT&Tag the authors should use important for the data in Figures 4 and 5 where many of the changes highlighted in the Genome Viewer snapshot do not appear particularly robust. Are these called as significant using rigorous comparisons that consider variability between replicates? (In fact, the authors comment that the tracks shown are averages of the replicates since there was variability.) This need for statistical measurements of differences is notable in many instances but is exemplified in

Figure 4D. The most robust changes in ATAC-seq and H3K27ac signal upon depletion of NR5A2 are in peaks that are lowly occupied by the factor. This is counterintuitive as one might predict that regions that depend most strongly on the factor would be those with the robust CUT&Tag signal. It is possible that these lower peaks, which also show the lowest signals for ATAC-seq and H3K27ac, are the most variable between replicates and are therefore the most subject to minor differences. Using more statistically robust means of calling differences and experiments to confirm that the differences observed are biological and not technical (such as spike in normalization, etc) is important.

We thank the reviewer for this critical suggestion, which was also made by reviewer 2 (comment 3). We have performed differential binding analysis by DiffBind as suggested. Our analysis identified 158 and 1,348 regions that showed decrease in H3K27ac upon *Nr5a2* KD and *Nr5a2 Klf5* KD conditions, respectively (new Figure 4C). Following this, we added a heatmap showing enrichment of H3K27ac on gain and loss of regions identified by differential binding analysis. We observed a more reduction of H3K27ac in at *Nr5a2 Klf5* KD at the NR5A2 and KLF5 co-occupied regions (new Figure 4D), which supports our claim.

The statistical power of differential binding analysis was not as powerful in our ATAC-seq datasets as there were only 2 replicates per conditions. We observed inconsistency of signals between the two replicates of ATAC-seq data in the non-targeting siRNA control conditions, which could partially explain the small number of affected regions detected in differential binding analysis of ATAC-seq dataset (Figure R5). Nevertheless, chromatin accessibility showed a similar trend in regions where H3K27ac is affected (new Fig. S5D), and a reproducible reduction of chromatin accessibility near down-regulated genes by *Nr5a2* KD is observed (new Figure S5F). The observations of decreased accessibility is consistent with previous studies (Lai et al., 2023; Festuccia et al., 2024). To ensure accuracy in data representation, we removed merged replicate ATAC-seq data from the analysis.

2. The explanation for the focus on the GATA and KLF families is not clear from the manuscript as written. Indeed, it would be important to note prior work in this context. For example, in Festuccia et al. Science 2024 it is suggested that KLF transcription factors might function with NR5A2. Similarly, the authors need to address other data from this manuscript that appears to contradict their study. In Figure 3 of Festuccia et al., GATA6 levels in E2.75 embryos that are mutant for maternal and zygotic NR5A2 appear to have normal levels of GATA6. This seems to contradict the data shown in Figure 5B where GATA6 levels are decreased upon RNAi targeting *Nr5a2*. The authors must address this discrepancy.

We thank the reviewer for seeking clarification on these points.

- We pursued GATA and KLF families based on a *de novo* motif search in NR5A2 peaks in our data. We described our rationale in lines 148-150.
- We apologize for unintentionally not mentioning prior work. We added the following text (lines 150-151) “KLF motifs had also been identified in *Nr5a2* peaks at the 8-cell stage previously (Festuccia et al., 2024).”
- Thank you for raising this discrepancy. Figure 3 of Festuccia et al. shows distinct cell stages (8-cells and 16-cells) in DHet (control) and maternal-zygotic *Nr5a2* knockout, which show similar abundance of GATA6. Whether these embryos show GATA6 abundance like in the wild-type or already at a reduced level is not known since no wild-type embryos were provided as a control. This makes it rather difficult to draw a direct comparison between the published data and our data. In addition, the previous study did not provide a quantification of GATA6 across cells, so a potential mild reduction in protein abundance could have been missed. We would also like to point out that our data is consistent with findings reported in Lai et al., Cell Research, 2023. Furthermore, the reduction of H3K27ac upon *Nr5a2* One possible explanation for this discrepancy is a distinct transcriptional response between genetic knockout and siRNA-mediated knockdown of NR5A2.

3. The data presented in Figure 6 G-I do not obviously support the claimed co-binding of NR5A2, KLF5 and GATA6 on the nucleosomes. The migration of the complexes on the gel is not clearly different when incubated with additional factors, and the decrease in free nucleosome is possibly explained by binding individual proteins. Are there conditions in which the resolution of the multiply

bound nucleosome could be better resolved? This would be important for supporting the claim.

We appreciate the reviewer's insightful comment. Indeed, the decrease in free nucleosomes involves binding of individual proteins. In this *in vitro* assay, approximately 50% of nucleosomes were bound by NR5A2 (Figure 6G, lane 2. Figure 6H, lane 2), and additional factors independently bound either to free nucleosome or to the NR5A2-nucleosome complex. The bands above the free nucleosome but below the NR5A2-nucleosome complex correspond to KLF5 DBD or GATA6 DBD-nucleosome complex (Figure 6G, lanes 3 and 4. Figure 6H, lanes 3-5). Therefore, the shifted band above the NR5A2-nucleosome complex corresponds to the NR5A2-nucleosome complex bound by KLF5 DBD or GATA6 DBD.

To improve the separation/migration of each complex, we purified KLF5 DBD and GATA6 DBD fused to maltose binding protein (MBP, 42kDa) (Figure R10A). We then performed an electrophoretic mobility shift assay with the exact same buffer conditions as before. We detected a supershift in the presence of additional factors, indicating the formation of higher-order complexes containing NR5A2, KLF5, and GATA6 on nucleosomes. (Figure R10B). However, the MBP-fused DBDs showed smeared bands, likely due to variable conformations of the MBP tag connected to the flexible linker. Therefore, we believe that the current data present the best resolution to separate each complex under the tested conditions, and the observed supershift supports the proposed co-binding of NR5A2, KLF5, and GATA6 on nucleosomes.

Figure R10: EMSA with MBP-fused DBD. (A) Protein purification of MBP-fused KLF5 DBD and MBP-fused GATA6 DBD. (B) B1 Nucleosomes (0.05 μ M) were incubated with NR5A2, MBP-KLF5 DBD, or MBP-GATA6 DBD at room temperature for 30 min in a reaction buffer (20 mM Tris-HCl (pH7.5), 120 mM NaCl, 1 mM MgCl₂, 10 μ M ZnCl₂, 1 mM DTT, 100 μ g/ml BSA). Nucleosomes were analyzed by 4.5% native-PAGE and detected by Alexa Fluor 647 fluorescence.

4. Some of the data are challenging to interpret given the complex interactions between these factors. The single and double knockdown experiments are nice but hard to interpret given the effect of Nr5a2 knockdown on KLF5 levels. The evidence that KLF5 and GATA6 regulate NR5A2 (as claimed on line 351) is not particularly convincing and would be strengthened by testing the binding of NR5A2 in the knockdown of KLF5 or GATA6.

Similarly, the redundancy between KLF and GATA family members makes simple conclusions difficult. We encourage the authors to tone down some of their conclusions. As an example, the heading on line 250 should be modified. The data show that KLF5 binds to the promoters of other family members and that their expression changes in the single knockdown (but notably not the double knockdown). As such, the data do not convincingly show that "KLF5 directly regulates other KLF family members." Similar arguments about direct regulation are made for NR5A2 on line 275 that should be modified to take into accounts that some of these effects may be indirect despite binding of the given factor. Statements in the abstract are also over interpreted. For example, the data supporting the cooperation of KLF5 with NR5A2 in H3K27ac is not very strong and therefore the statement on line 26 that "KLF5 cooperates with NR5A2 to enhance H3K27ac deposition" should be

modified.

We thank the reviewer for these valuable points.

- We thank the reviewer for this critical suggestion, which was also made by reviewer 2 (comment 5).

We examined the NR5A2 binding profile upon *Klf5* KD. To increase the sensitivity of chromatin binding profiles with a limited number of input cells, NR5A2 Targeted insertion of promoters sequencing (TIP-seq) (Bartlett et al., 2021) was performed (control 8-cell embryos, rep1: 25, rep2: 26. *Klf5* KD 8-cell embryos, rep1: 30, rep2: 30). This experiment showed that NR5A2 occupancy is not significantly altered upon *Klf5* KD, suggesting that NR5A2 binding is largely independent of KLF5 binding (new Figure S6B,C). Because *Gata6* KD was inefficient up to the 8-cell stage, we could not test whether NR5A2 chromatin binding depends on GATA6.

- Based on your comments and those of Reviewer 1, we removed the section “KLF5 directly regulates other KLF family members”.
- We apologize for any instances of over-interpretation of the data. We have extensively revised the manuscript and toned down our conclusions.

Other critiques:

1. Clarity would be improved if the authors were more explicit about how hours post fertilization correspond to the various stages (2C, 4C, 8C, morula). Similarly, in the introduction it would be useful to connect the morula stage to the blastocyst and the differentiation of ICM and TE.

- We added the information on the timing of collection of embryos in the legend of Figure 1 (lines 458-459). We also provided it in the methods section under “Collection of mouse embryos” (lines 589-591).
- We have amended the introduction according to your suggestion (lines 49-53).

2. Figure 1D is surprising. It appears that most of the NR5A2 binding at SINE B1 elements changes between stages with ~70% gained and lost between the 2C-4C and over 50% between the 4C-8C. The authors focus on the gain (line 113), but this seems misplaced as there are nearly equally dramatic changes in gains and losses between stages.

We thank the reviewer for the insightful comment. We have added a modified figure showing gain and loss peaks along with the proportion of *B1/Alu* (new figure 1E). Although our main conclusion remains unchanged, this analysis provides a more comprehensive view of the stage-specific transition in NR5A2 binding at *SINE B1* elements. NR5A2 peaks at *SINE B1/Alu* were highly gained at the 2-cell-to-4-cell and 4-cell-to-8-cell transition (new Figure 1E). In contrast, in the 8-cell-to-morula transition, the gain of NR5A2 peak at *SINE B1/Alu* was decreased, accompanied by a pronounced loss of peaks at *SINE B1/Alu* (new Figure 1E).

3. The authors make a strong claim concerning GO term analysis and the function of cluster 3 in stem cell maintenance in the results. This text would be better suited for the discussion.

The category of stem cell maintenance in cluster 3 includes ICM genes, such as *Nanog* and *Oct4*. We refer to the GO term analysis at the beginning of the discussion (lines 352-364).

4. Can the authors define hatching as discussed on line 193?

We defined hatching as “breaking out of zona pellucida” (line 200)

5. On line 303, the authors might substitute “in principle” for “principally”. The first suggests the possibility. The second makes it sound like this is the major form of binding.

The reviewer is absolutely correct. We have amended the text accordingly (line 306).

6. In the legend for Figure 2A, please specify to what the comparison was made to identify enrichment.

We apologize for the lack of clarity. The color indicates odds ratio between observed and expected

motif occurrence within the peak. And, the size of circle indicates the p-value. This has been added to the figure legend (lines 474-476).

7. In the Figure 4 legend - the second 'D' (in line 490) should be 'E'.

We apologize for our typo and corrected it.

8. The model in Figure 7B would be strengthened if the authors related the molecular events to developmental timepoints.

This is a very good suggestion. We incorporated the molecular events to developmental time points in Figure 7B.

Second decision letter

MS ID#: dev.205059R1

MS TITLE: Feed-forward loops by NR5A2 ensure robust gene activation during pre-implantation development

AUTHORS: Wataru Kobayashi; Siwat Ruangroengkulrith; Eda Nur Arslantas; Adarsh Mohanan; Kikue Tachibana

ARTICLE TYPE: Research Article

Dear Dr Tachibana,

I am happy to tell you that your manuscript has been accepted for publication in Development, pending our standard publication integrity checks.

Reviewer 1

The authors have addressed all my concerns, either by additional experiments or by clarifying existing data, and have substantially improved the manuscript. I recommend for publication in Development.

Reviewer 2

Authors went beyond what I requested and now the study satisfies all prior concerns.

Reviewer 3

The authors have adequately addressed all of our comments and removed data that was not robust. We support publication in Development.